



# Cost Effective Off-Grid Automatic Precipitation Samplers for Pollutant and Biogeochemical Atmospheric Deposition

Alessia A. Colussi[1], Daniel Persaud[1], Melodie Lao[1], Bryan K. Place[2,‡], Rachel F. Hems[2,§], Susan E. Ziegler[3], Kate A. Edwards[4,‡], Cora J. Young[1,2], and Trevor C. VandenBoer[1,3]

[1] *Department of Chemistry, York University, Toronto, ON*
[2] *Department of Chemistry, Memorial University, St. John's, NL*
[3] *Department of Earth Science, Memorial University, St. John's, NL*
[4] *Canadian Forest Service, Natural Resources Canada, Corner Brook, NL*

‡ Now at: *SciGlob Instruments & Services LLC, Columbia, MD, USA*
§ Now at: *Department of Chemistry and Biochemistry, Oberlin College and Conservatory, OH, USA*
‡ Now at*: Climate Change Impacts and Adaptation Division, Lands and Minerals Sector, Natural Resources Canada, Ottawa, ON*

**Correspondence:** Trevor VandenBoer (tvandenb@yorku.ca)

## Abstract

An important transport process for particles and gases from the atmosphere to aquatic and terrestrial environments is through dry and wet deposition. An open-source, modular, off-grid, and affordable instrument that can automatically collect wet deposition samples allows for more extensive deployment of deposition samplers in fieldwork and would enable more comprehensive monitoring of remote locations. Precipitation events selectively sampled using a conductivity sensor powered by a battery-based supply are central to off-grid capabilities. The prevalence of conductive precipitation, initially containing high solute levels and progressing through trace level concentrations to ultrapure water in full atmospheric washout, depends on the sampling location but is ubiquitous. This property is exploited here to trigger an electric motor via limit switches to open and close a lid resting over a funnel opening. The motors are operated via a custom-built and modular digital logic control board, which have low energy demands. All components, their design and rationale, and assembly are provided for community use. The modularity of the control board allows operation of up to six independent wet deposition units, such that replicate measurements



(e.g., canopy throughfall) or different collection materials for various targeted pollutants can be
implemented as necessary.
We demonstrate that these platforms are capable of continuous operation off-grid for integrated
monthly and bimonthly collections performed across the Newfoundland and Labrador Boreal
Ecosystem Latitudinal Transect (47° to 53° N) during the growing seasons of 2015 and 2016.
System performance was assessed through measured power consumption from 115 volts of
alternating current (VAC; grid power) or 12 volts of direct current from battery supplies during
operation under both standby (40 or 230 mA, respectively) and in-use (78 or 300 mA, respectively)
conditions. In the field, one set of triplicate samplers was deployed in the open to collect incident
precipitation (open fall) while another set was deployed under the experimental forest canopy
(throughfall). The proof-of-concept systems were validated with basic measurements of rainwater
chemistry including: i) pH ranging from 4.14 to 5.71 in incident open fall rainwater; ii)
conductivity ranging from 21 to 166 uS/cm; and iii) dissolved organic carbon (DOC)
concentrations in open fall and canopy throughfall of $16 \pm 10$ mg/L and $22 \pm 12$ mg/L, respectively;
with incident fluxes spanning 600 to 4200 mg C m$^{-2}$ a$^{-1}$ across the transect. Ultimately, this
demonstrates that the customized precipitation sampling design of this new platform enables more
universal accessibility of deposition samples to the atmospheric observation community – for
example, those who have made community calls for targeting biogeochemical budgets and/or
contaminants of emerging concern in sensitive and remote regions.

**1.0    Introduction**
Atmospheric deposition is the central loss process for particles and gases to terrestrial and
aquatic surfaces (Pacyna, 2008). Particles and gases can be deposited by both dry and wet
deposition processes. Dry deposition is facilitated by the direct interaction of gases and particles
with boundary layer surfaces such as water, vegetation, and/or soil, while wet deposition involves
in-cloud scavenging and below-cloud interception of gases and aerosols by, e.g., rain droplets and
snow crystals (Fowler, 1980; Lovett and Kinsman, 1990). Dry and wet deposition are global
processes coupled to regional synoptic scale conditions, but their relative importance depends on
local sources and global transport of atmospheric analytes of interest. Dry deposition consists of a
variety of mechanisms for particles and gases, with fine mode particles (compared to ultrafine and
coarse mode particles) and their chemical constituents being more likely  to undergo atmospheric



long-range transport before eventually being deposited (Farmer et al., 2021). Wet deposition
occurs when such long-lived atmospheric particles and gases are included and/or scavenged into
cloud water and transported to the surface of the Earth in precipitation (e.g., snow and rain). With
the size and number of droplets in the atmosphere largely controlling the rate, wet deposition
depends on a variety of meteorological factors affecting precipitation, such as the size distribution
and concentration of ice and droplet nucleating particles, as well as the solubility, concentration,
and reactivity of gases (Lovett, 1994).
Ultimately, deposition plays an important role in pollutant distribution and biogeochemical
cycling of major nutrients, including long-studied nitrogen and sulfur in acid rain, alongside those
with increasing recognition of importance such as dissolved organic carbon (DOC) (Vet et al.,
2014; Safieddine and Heald, 2017). If high amounts of atmospheric pollutants or nutrient-bearing
compounds are deposited at an environmental interface, this could result in these compounds
contaminating the receptor site or exceeding their critical loads for organism or ecosystem
function, respectively (Meteorological Service of Canada, 2005; Clark et al., 2018; United States
Environmental Protection Agency, 2020). For example, the deposition of nitrogenous compounds
to ecosystems can be both desirable or undesirable (Zhu et al., 2015; Kanakidou et al., 2016;
Midolo et al., 2019). Although nitrogen deposition could be an additional source of nutrients to
plants, in excess it can result in eutrophication and oxygen deficiency in waterbodies due to algal
blooms (Pacyna, 2008). Not all atmospheric trace gases can be removed effectively via
precipitation, such as sulfur dioxide ($SO_2$) and nitrogen oxides ($NO_x$). When these are emitted to
the troposphere, they are instead transformed into acids which are readily incorporated into cloud
droplets. The nitric and sulfuric acid products are then deposited as environmentally harmful acid
rain. These same two classes of acid precursors are released by natural processes, but
anthropogenic processes (e.g., the burning of fossil fuels) have been long-established to drive past
precipitation acidity (Mohnen, 1988) alongside the subsequent policies enacted to mitigate them.
Recognizing the significance of atmospheric trace chemical deposition has led to the
establishment of monitoring networks aiming to provide critical data on their spatial and temporal
patterns of wet and dry deposition. Long-term wet deposition monitoring networks, like the
Canadian Air and Precipitation Monitoring Network (CAPMoN) and the National Atmospheric
Deposition Program (NADP), have enhanced scientific knowledge of temporal and spatial
deposition. As a result, this has allowed for the estimation of regional and continental deposition



rates of species regulated by national or international policies (Lovett, 1994). Data from these
networks have been critical to understanding the efficacy of policy to reduce environmental issues
like acid rain (Likens and Butler, 2020). In addition to implementing deposition networks, the
international community has created protocols (e.g., the Oslo and Geneva protocols) which have
achieved an 80% decrease in both North American and European $SO_2$ emissions since 1980
(Grennfelt et al., 2020). However, reductions in acid deposition have had unexpected slow
recovery in ecosystems leaving them sensitized – indicating a need for continued deposition
monitoring (Stoddard et al., 1999; Kuylenstierna et al., 2001).
While bulk deposition collection (i.e., a collection bucket or jug fitted with a funnel open
at all times; Hall, 1985) is both simple and economically feasible, this sampling method is subject
to bias through collection of inputs other than atmospheric deposition (e.g., bird droppings, insects,
plant debris). As a result, bulk collectors can overestimate total deposition and underestimate wet
deposition in a variety of locations (Lindberg et al., 1986; Richter and Lindberg, 1988; Stedman
et al., 1990). Although it is a more costly and time intensive method when compared to bulk
deposition collection, the major appeal of measuring isolated wet deposition is that it can be
conducted so long as one has a container made of a suitable material for the target analytes and is
able to time its deployment and collection to isolate the wet deposition event. Further innovation
can reduce bias and improve the preservation of samples, such as the use of sensors to automate
isolation of collected precipitation or the addition of polymeric mesh barriers to reduce debris input
in windy environments (Lovett, 1994) - yet commercial solutions often come at a substantial
expense.
Modern emerging issues that require the continuation of existing deposition measurements
or expansion of observation programs revolve around identifying and quantifying compound
classes of concern, such as persistent organic pollutants (POPs). This group of compounds are
characterized by their persistence within the environment and organisms, a high tendency to
bioaccumulate, as well as an ability to exert toxic effects – with various POPs being classified as
carcinogenic (Safe, 2003; Van den Berg et al., 2006). As a result, atmospheric deposition is an
important means by which POPs, or their precursors, are transported from an emission source back
to surface environments (Gregor and Gummer, 1989; Fingler et al., 1994; Pickard et al., 2018).
The deposition of POPs (e.g., polybrominated diphenyl ethers, polychlorinated biphenyls,
polycyclic aromatic hydrocarbons, etc.) can be monitored using suitable collectors made of amber-



coloured glass or stainless steel (Fingler et al., 1994; Amodio et al., 2014). For example, on grid
bulk collectors and wet-only collectors, with lids triggered by detected conductive precipitation,
have been used to isolate the relative role of dry and wet deposition processes (Pekey et al., 2007).

When targeting biogeochemically relevant species in deposition collectors, additional

standard practices have been developed to improve the representativeness of sample composition.
First, an appropriate monitoring site must be selected. Three categories of siting criteria,
established by organizations such as CAPMoN and the NADP, are of particular importance: (i)
site representativeness and physical characteristics, (ii) distance from potential pollution sources,
and (iii) operational requirements (Canadian Air and Precipitation Monitoring Network, 1985;
National Atmospheric Deposition Program, 2009). This means that each site must be a location
that receives precipitation representative of the hydrologic area; is ideally not within 500 m of
local pollution sources, such as wood-burning stoves, garbage dumps, and vehicle parking lots;
and is accessible for daily collections, maintenance, and be serviced by reliable 115 VAC electrical
power (Canadian Air and Precipitation Monitoring Network, 1985; National Atmospheric
Deposition Program, 2009). Despite these guidelines, there are many reasonable scenarios in
which these siting conditions cannot be met. As an example, remote sample collections are often
required for global assessments on persistent contaminants or nutrients of biogeochemical
importance. Remote locations, however, can result in sampling sites with no power provision,
infrequent sample collection, and/or the infrastructure-bearing location itself is a source of the
targeted pollutants. As a result, innovation in collection strategies such as time-integrated off-grid
sampling, with modularity in the deployment of replicates, as well as materials for quantitative
collection of environmental targets, is still needed to expand and/or modify networks to meet future
monitoring and policy needs.

The lack of organic nitrogen (ON) measurements within universally established sampling

and measurement procedures serves as a general example of the substantial knowledge gaps that
may result when translating limited data sets to the wider global picture. This includes incomplete
speciation and quantification across precipitation, aerosol, and gas phases. Monitoring systems
that support U.S. deposition assessments (e.g., the NADP) only characterize the inorganic fraction
of wet deposition. This results in an incomplete assessment of organic compounds, including
several atmospheric groups that have important environmental effects, such as the above-
mentioned POPs, reactive nitrogen ($N_r$), and reactive organic carbon (ROC). Contributions to total


$N_r$ by ON compounds in precipitation within the U.S. have been estimated to be between 3% to
33% (Benedict et al., 2013; Chen et al., 2018), with a demonstrated abundance of unexpected
organic compounds by high resolution mass spectrometry (Altieri et al., 2009, 2012; Ditto et al.,
2020). There is a clear and substantial need to improve knowledge of the $N_r$ nutrient pool, amongst
other compound classes, to correctly estimate the magnitude of exchange between the atmosphere
and terrestrial and/or aquatic environments.

In biogeochemical cycles, improvement of constraints in atmospheric carbon linkages to

terrestrial and aquatic processes are also critical to correctly assess climate feedbacks and reduce
uncertainty in Earth system models. Measurement of atmospheric DOC transport to surfaces has
been limited and impedes landscape scale carbon balance from being obtained (Casas-Ruiz et al.,
2023). The pool of compounds from which DOC is derived in the atmosphere has also been limited
and is only recently seeing an increase in research intensity. Reactive organic carbon, defined as
the sum of nonmethane organic gases and primary and secondary organic aerosols (Safieddine and
Heald, 2017), is an important driver of oxidative chemistry within the atmosphere. The major
removal mechanism of water-soluble organic compounds produced through oxidation from the
atmosphere is by dry deposition of particle-bound pollutants and scavenging by rainfall (Jurado et
al., 2004, 2005). When ROC is scavenged by dry deposition or rainfall, it becomes DOC and enters
terrestrial and aquatic systems. This concept has generated increasing attention around the controls
on and composition of DOC in deposition samples. The primary interest is in getting mass closure
on atmospheric ROC. Deposition measurements of ROC compounds are also needed since they
play a crucial role in the formation of a plethora of secondary species: ozone ($O_3$), particulate
matter, and carbon dioxide ($CO_2$) (Safieddine and Heald, 2017; Heald and Kroll, 2020).

There are several evolving drivers around studying atmospheric ROC; for example, light-

absorbing organic carbon that accumulates in the atmosphere can affect global radiative balance
and can change over time through photochemical transformations in the condensed phase (Saleh,
2020; Wang et al., 2021; Washenfelder et al., 2022; Geroge, 2023).  Reactive organic carbon can
also influence cloud formation and its contribution to precipitation acidity (Avery et al., 2006;
Ramanathan and Carmichael, 2008).  Measurements of speciated ROC are difficult due to the
chemical complexity of emitted compounds and oxidation products (Heald and Kroll, 2020). To
circumvent this, monitoring and quantifying DOC can be used as a proxy to estimate the total ROC
in precipitation. However, quantitative measurements of DOC in precipitation samples are sparse



due to its relatively low concentration which has been reported between 0.1 to 10 mg C $L^{-1}$ in
incident precipitation (Iavorivska et al., 2016; Safieddine and Heald, 2017). Recently, calls for
carbon closure on atmospheric processing of ROC make this measurement of increasing
importance (Kroll et al., 2011; Heald et al., 2020; Barber and Kroll, 2021). Similarly, to obtain net
landscape or watershed carbon exchange, studies require effective methods for capturing and
preserving atmospheric DOC deposition to constrain biogeochemical linkages at global interfaces
as outlined above.

In this work, we present the design of a custom-built automated array of precipitation

samplers that can be operated both on- and off-grid for wet deposition collection. The purpose of
these samplers is to enable cost-effective collection of monthly integrated water-soluble
conductive atmospheric constituents deposited in remote environments without grid power or
routine access. A sensor interfaces with a custom-built motor control board capable of operating
up to six independent wet deposition units such that canopy throughfall (TF) and incident
precipitation measurements are possible to collect in replicate. The materials used can be easily
changed in order to optimize collection of a wide array of target analytes, such as POPs or DOC.
We demonstrate that these platforms are capable of continuous operation off-grid for monthly wet
deposition collection of precipitation across the Newfoundland and Labrador Boreal Ecosystem
Latitudinal Transect (NL-BELT) during snow-free periods in 2015 and 2016. Extremes in system
performance were evaluated by testing the power consumption of a sampling array from spring
through fall when paired with a solar top-up system, and during snow-free winter conditions using
only a battery. The two years of field samples were collected using an array of six collection units,
with triplicate collection of both incident precipitation and throughfall from rain passing through
a forest canopy. Samples were analyzed in terms of deposition volumes relative to total bulk
volumes, to assess the reproducibility of replicate samples, and determine the fraction of
conductive rainfall collected from the total volume of precipitation. The captured fraction
compared to total volume deposited is used to gain insight into limiting analyte dilution effects
and improving deposition method detection limits. Chemical parameters of pH, conductivity, and
DOC fluxes were then used to validate this proof-of-concept system. Measurements of pH and
conductivity for rainwater are very well-established in the literature and serve as a baseline
reference to ensure that the samples collected by the new devices presented in this work are
consistent with what is expected in samples from a remote coastal environment, given the selective



sampling strategy. We then move away from these well-established parameters to quantify DOC
fluxes to demonstrate the potential of these samplers in application to automated collection of
analytes of emerging importance and interest in remote locations spanning our latitudinal transect.

**2.0 Materials and Methods**
**2.1 Precipitation Sampling Array Components**

Each automated precipitation sampling setup can be operated as an array, here being used

in groups of up to six collection units (Figure 1). A collection unit is a simple opaque doored box.
The box protects the sample containers against exposure to direct sunlight and provides a mounting
location for the funnel and lid, while also facilitating easy exchange of sample containers. The
collection units can be fitted with stabilizing legs that allow them to be bolted to concrete or pinned
by retaining rods when on soil. In both cases, this prevents tipping and loss of sample during high
winds or wildlife-sampler interactions (e.g., Figures 2 and S1). The collection of precipitation is
facilitated by a funnel mounted through the top of the sampling unit. The funnel tip extends into
the opening of the sample collection container placed inside. The connection can be sealed to better
preserve volatile analytes with tubing that passes through a sealed grommet (P/N 9280K34,
McMaster-Carr) to enter the sample collection container and minimize evaporative losses.
Precipitation events are sampled selectively by modulating the position of a lid over the funnel
with an electric motor. The collection unit motors are operated by a digital control board, which
interfaces with a precipitation sensor and requires 12 volts of direct current (VDC) power supplied
to this system. Switches detecting the lid position ensure complete opening or closure of the funnel
mouth for each collection unit.

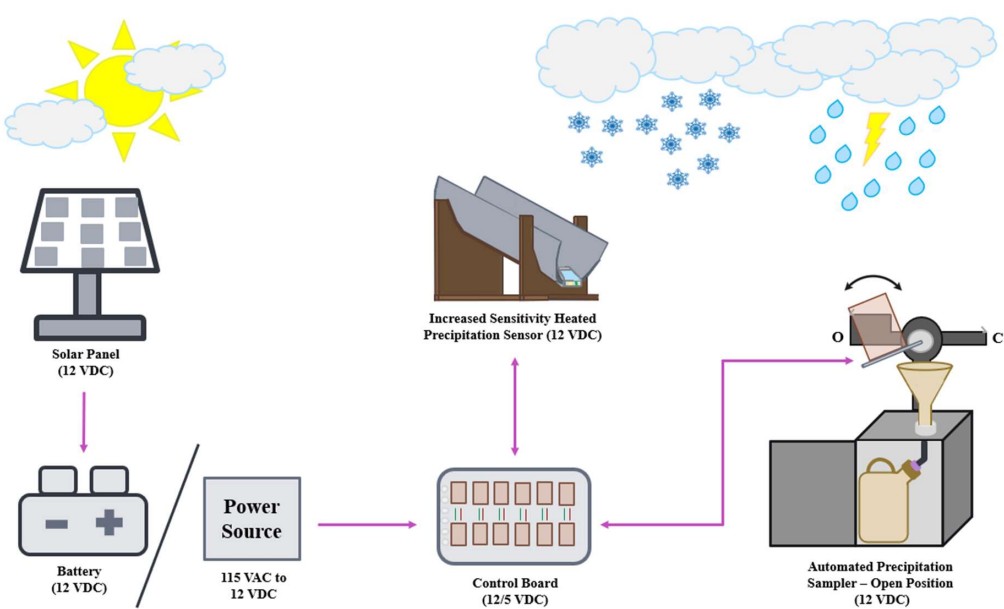

**Figure 1.** Schematic of custom-built automated precipitation sampling array components for off-
grid wet deposition collection. The pink arrows denote the direction of electrical signal and power
exchanged between components. The curved black arrow indicates the rotation of a motorized lid
to obtain open (O) or closed (C) sampler configurations.

**2.1.1 Collection Units**
The collection unit materials to date have been made of both 3/8" plywood and black
polyacrylate sheeting. The materials have demonstrated high durability on the order of four years
under field conditions (Figures S1 and S2). Opaque materials were explicitly selected to minimize
photochemical reactions and growth of photosynthetic microorganisms within the sample. The
dimensions of the collection unit are detailed in Figure 2. Each can accommodate sample
containers up to 20 L in volume for collection in locations with large monthly wet deposition
volumes, such as in Newfoundland and Labrador (Table 1).

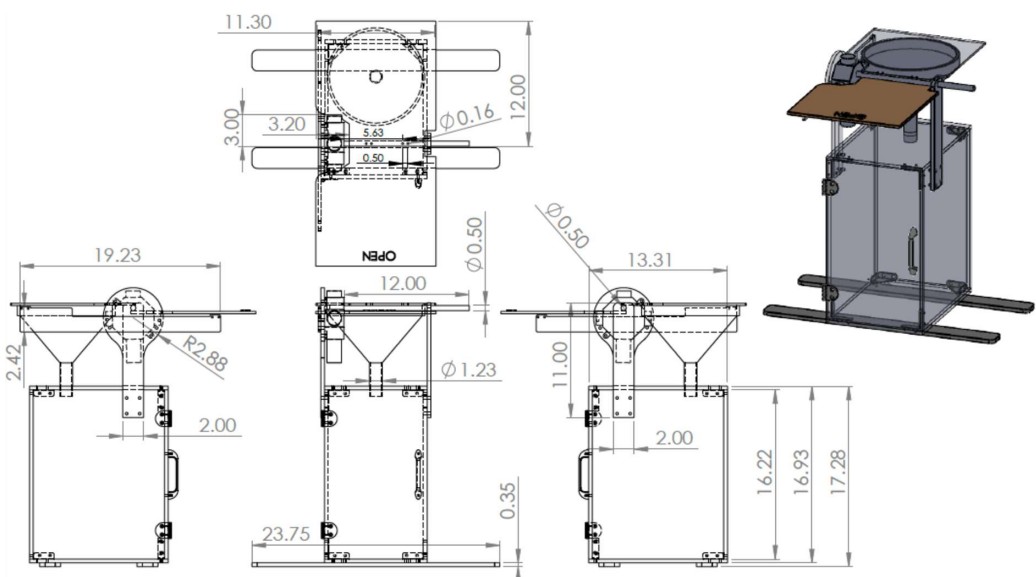

**Figure 2.** Detailed collection unit schematic with all dimensions provided in inches. Further specifications for the lid dimensions can be found in Figure S3. The shaded 3D rendering depicts both open and closed states for the lid, positioning of legs to secure it to surfaces, placement of corner brackets, and the door handle and hinges.

The box panels can be joined using hardware inserts (P/N 1556A54 and 1088A31, McMaster-Carr, Aurora, OH), 3D printed corners (Figure S4), or along the box edges with screws if using wood. The door is attached with two hinges (P/N 1549A57, McMaster-Carr) and held closed with a magnetic contact (P/N1674A61; McMaster-Carr) or hooked latch. The electric motor controlling the lid is enclosed in a standard polyvinylchloride electrical junction box, which is attached to a short paddle mounted on one side of the collection unit. Here we used an electric worm-gear motor (12 VDC, 2 rpm; TS-32GZ370-1650; Tsiny Motor Industrial Co., Dong Guan, China) mounted inside the enclosure with matching hex bolts (P/N 91251A146, McMaster-Carr) that passed through the weather-tight cover while the drive shaft protrudes through a 3/8" hole drilled in the cover. The drive shaft has a flat edge to affix the lid rod using a short set screw (Figure S5) that is cemented semi-permanently in place with thread locking compound (P/N 91458A112; Loctite Threadlocker Blue 242; McMaster-Carr). The lid rod is 3/8" aluminum machined on one end to allow connection to the motor drive shaft (Figure S5) with four threaded holes along its length to affix the lid (Figure S3). The lid rod passes through a second mounting





paddle on the box that keeps the lid level and capable of isolating the funnel from the atmosphere
in the absence of precipitation. The lids used here were made of 1/8" Lexan polycarbonate sheet.

Selective precipitation sampling is performed using a logic-based assessment of sensor and

switch states (defined in Figure S6) by the control board quadNOR gate chip (Fairchild
Semiconductor P/N DM74LS02) which activates the H-bridge motor driver chipset (Figure S7).
A 12 VDC signal drives the clockwise or counterclockwise rotation of the motor, installed in a
suitable port of the junction box, via a cable from the control board, which passes through a
weather-tight compression fitting (e.g., Home Depot SKU# 1000116446). The motor rotation
signal is interrupted when the lid makes contact with one of two weather-tight limit switches (P/N
SW1257-ND; Omron, Digi-Key Electronics, Thief River Falls, MN) mounted on opposite ends of
a horizontal armature connected to the vertical motor mounting paddle (Figure 2). The switches
controlling the lid location ensure that the funnel is completely open or covered as necessary for
precipitation collection. The funnels used in this work are 20 cm in diameter and made from high-
density polyethylene (HDPE; Dynalon, P/N 71070-020, VWR International, Mississauga, ON). A
7" x 5" piece of filtration mesh (P/N 9265T49, McMaster-Carr) that was tied together as a fitted
cone insert with Nylon thread (e.g., fishing line) to prevent large debris entering the sampler
containers when used, for example, in the collection of TF precipitation under a forest canopy
when accompanying litterfall is also expected. The exit of the funnel directs the collected
precipitation into the narrow-mouth opening the container inside the collection unit, such as 20 L
HDPE jugs or 10 L HDPE jerricans (Bel-Art Products; P/N 11215-314, VWR International).

### 2.1.2 Heated Precipitation Sensor

The detection of rain modulates the opening and closing of the collection units by an

interdigitated resistive sensor (M152; Kemo Electronic GmbH, Geestland, Germany; Figures S6
to S8). The rain sensor detects conductive deposition by the completion of a conductive circuit
when electrolytes bridge the connection between the interdigitated gold electrodes. The sensor is
supplied with 12 VDC from the power system to trigger a relay when precipitation conductance
above 1 MΩ·cm conductivity is detected (determined experimentally, see Sect. S1). An output of
12 VDC is sent to the digital control board by the relay when rain is sensed, or 0 VDC in its
absence, for signal processing and motor control (Figure S7). To increase the sensitivity of this
sensor and to extend the sampling duration when conductive atmospheric constituents are



completely washed out of the atmosphere, a sloped tin chute (e.g., Home Depot SKU# 1001110514) was added to extend the surface of the rain sensor. The sensor was placed at the end of the chute and sealed in place with caulking to allow water droplets to move easily from the chute onto the sensor.

The angle of the chute can be adjusted to control the momentum of collected droplets so that they collect on the sensor surface and only flow off it when the rate of precipitation exceeds the sensor evaporation capability. When soil is available, two bent rods can be used to hold the chute at the optimized angle of 10° (Figure S2). They are inserted into the soil and the chute is affixed to the tops of the rods with zip ties passed through small holes drilled in the sides of the chute, which are subsequently sealed with caulking. When soil is unavailable, for example in urban environments, we have created a mounting frame to hold the chute at the optimized angle of 10° (Figure S8). When precipitation is detected the sensor surface draws current up to 1.0 ampere (A) into a heater to actively evaporate water from its surface so that it accurately detects the active period of rain events. The heated sensor has undergone preliminary field tests and is also capable of detecting ice and snow, provided they contain electrolytes.

### 2.1.3 Power Supply Systems

Power for this system can be supplied from a battery at 12 VDC or using a 115 VAC to 12 VDC transformer power supply (P/N 285-1818-ND; TDK-Lambda Americas, Digi-Key Electronics). Depending on the duration of sampling and the time of year, the battery capacity can be changed to suit power needs (Sect. 3.2.2). To provide sufficient power density in this study, over one to two month-long collection periods, the battery capacity was carefully matched; with top-up options implemented when prolonged or high-frequency precipitation was expected. Absorbent glass mat (AGM) marine deep cycle batteries can withstand discharge events down to less than 60 % capacity and are robust under nearly all environmentally relevant temperatures (≤ -20 °C to 40 °C). Additionally, these batteries interface easily with solar charging options as they are able to accept high current input. Monthly collections in Newfoundland were powered with 76 amp-hour (Ah) AGM batteries (Motomaster Nautilus; Ultra XD Group 24 High-Performance AGM Deep Cycle Battery, 12 VDC) topped up by a 40 W solar panel interfaced with a charge controller to prevent overcharging (Coleman; Model # 51840, max current of 8 A at 14 VDC).





For collections made every second month in Labrador, a 120 Ah battery with the same

solar top-up strategy was used to ensure continuous operation. For either remote field deployment,
batteries and charge controllers were housed in a Pelican™ case (Model 1440, Ocean Case Co.
Ltd., Enfield, NS) fitted with weathertight bulkhead cord grips (P/N 7529K655, McMaster-Carr)
through which charging and power cables were passed (Belden, Coleman; S/N 7004608,
70875227, Allied Electronics, Inc., Ottawa, ON). Humidity in all weatherproof cases was
minimized by exchanging reusable desiccant packs (Ocean Case Co. Ltd.) when depleted batteries
were exchanged for fully charged replacements. Solar panels were repositioned monthly to
optimize orientation for solar power provision. Using either power source, the control board
converts and distributes the 12 VDC to the other components in the precipitation sampling array.

### 2.1.4 Custom Control Board

A custom control board to operate a six-collection unit array was designed based on prior

digital logic circuits for standalone collectors (VandenBoer, 2009). The 12 VDC battery or
transformer output is supplied directly to the rain sensor and relay, as well as to the motor drivers
for lid opening (Figure S9). Each collection unit is controlled independently to ensure lids are fully
opened or closed, thereby requiring six replicate motor driver control circuits that respond to their
independent switch signals. The remainder of the signaling and digital logic operates on 5 VDC
which is produced by on-board voltage regulators (Micro Commercial Co; P/N MC7805CT-BP,
Digi-Key Electronics). The lid switches are provided with 0 and 5 VDC to indicate collection unit
open or closed status (Omron Electronics; P/N D2FW-G271M(D), Digi-Key Electronics). The
signals from the sensor and switches connect to the board through four-conductor cable (Belden;
S/N 70003678, Allied Electronics Inc.) passed through weathertight bulkhead cord grips and
secured to screw terminals (Figure S9). The sensor and switch signal inputs interface with a quad
NOR GATE chipset (Texas Instruments; P/N 296-33594-5-ND, Digi-Key Electronics) to trigger
the motor driver (STMicroelectronics; P/N 497-1395-5-ND, Digi-Key Electronics) such that it
rotates or remains stationary. The additional resistors, capacitors, and diodes are necessary to
maintain stable signaling throughout the printed circuit board (Figure S9, Table S1).

The custom control board was housed in a Pelican™ case (1400 NF; Pelican Zone,

Mississauga, ON) fitted with cut-to-use foam inserts and a reusable desiccant pack that was also
exchanged alongside those for the battery cases. All collection units, sensors, and power supply


cables were passed through eight weathertight bulkhead cord grips and fixed to screw terminals
on the board. The opposing ends of the cables were fitted with weathertight Bulgin Buccaneer 400
or 4000 Series circular cable connectors (Table S2; Allied Electronics, Inc.) to allow easy field
installation with mated connectors on the cables originating from each of the previously mentioned
array components. Connected cables could then be buried in shallow soil trenches to reduce the
attention of gnawing animals, as well as potential entanglement hazards with other wildlife.
Precipitation events were logged from the control boards using a HOBO 4-channel analog data
logger (UX120-006M; Onset®, Bourne, MA) that records the sensor, switch, and motor voltages.
The fourth channel is reserved to monitor battery or power supply voltages over time (Sect. 3.2).

**2.2 Power Demand and Management Tests**

Power demand was calculated based on the cumulative component requirements prior to

the selection of batteries. This was to ensure adequate capacity to collect samples over one to two
month-long field deployments and are sufficient for an assumed worst-case scenario of one week
of constant rain without solar power charge restoration. Solar panel power production capacity
was determined based on the calculated energy required to recharge the battery. As a result, we
selected the 40 W panel which could complete charging at 14 VDC with a week of direct sunlight
at 8 hours per day. The power demand for a six-sampler array was measured in standby and during
operation with a digital power meter (Nashone PM90, Dalang Town, China) in real-time when
supplying 12 VDC with a transformer. Contrasting power demand tests were performed under
different environmental conditions and power management configurations. The first was
performed using the 76 Ah AGM battery with a solar top-up in an urban environment from July
through August 2018, while the other was performed using a 103 Ah AGM battery alone from
January through February 2019.

**2.3 Continuous Monthly Collection of Remote Samples at NL-BELT**

Four arrays of six automated collection units and total deposition samplers were deployed

within one forested experimental field site located in each of the four watershed regions of the NL-
BELT between 2015 and 2016 (Table 1, Figure S10). The watersheds span 5.5° latitude from the
southernmost site Grand Codroy (GC), through the colocated Pynn's Brook (PB) and Humber
River Camp 10 (HR) sites, to Salmon River (SR) as the highest latitude site on the island of



Newfoundland. The northernmost forested watershed, Eagle River (ER), is located in southern
Labrador and extensive details characterizing each of the four sites can be found in Ziegler et al.
(2017). All sampling locations are far from anthropogenic pollutant point sources, except for the
ubiquitous presence of marine sea spray from the nearby marine coastlines. The total deposition
samplers were identical to the automatic collection units except that they were not fitted with a
motor arm and lid, so they did not require a source of power. Three of the six automated samplers
were deployed in the open at a distance from the forest stand, equal to or greater than the height
of the trees, in line with CAPMoN and NADP guidelines. The other three automated samplers
were placed under the canopy to collect TF precipitation within the forest sites. These samplers
actively collected wet deposition into integrated monthly (Newfoundland) or two month
(Labrador) samples during snow-free periods (approximately June through November). The arrays
were collected and stored during the winter months while total deposition samplers remained in
field locations year-round. It is also important to note that during the growing season, sample
collections were made at the same time – that is, open fall (OF) and TF deposition were collected
on a single day at each sampling site and within a few days of each other across the transect.
Collected sample volumes were compared between the automated samplers and total deposition
collectors for each collection interval as a check on proper function (i.e., less than or equal volumes
in automated samples).
**Table 1.** NL-BELT sampling site details provide locations and identifiers, alongside those from
long-term weather stations operated by Environment and Climate Change Canada (ECCC). Soil
pH was determined from samples collected at the same time as precipitation. Mean annual
temperature was derived from ECCC climate normals. Annual total deposition precipitation
volumes were either measured for the 2015-16 period (ECCC, This Work) or calculated by the
Oak Ridge National Lab DAYMET archive.

| Sampling Site | Sampling Site Location | Station (Climate ID) | Station Location | Soil pH[a] | MAT (°C)[b] | Average Annual Precipitation (L) | | |
|---|---|---|---|---|---|---|---|---|
| | | | | | | ECCC[c] | DAYMET[g] | This Work |
| Grand Codroy (GC) | 47°50'43.1"N 59°16'16.0"W | Stephenville A (8403801) | 48°32'29.00" N 58°33'00" W | 3 to 4 | 5.0[c] | 53.2 | 58.9 | 45.6 (+5.17) |
| Pynn's Brook (PB) | 49° 05' 13.20"N 57° 32' 27.60" W | South Brook Pasadena (8403693) | 49°01'00" N 57°37'00" W | 3 to 4 | 4.6[c] | 21.4 | 54.3 | 38.6[h] |
| Salmon River | 51°15'21.6"N 56°08'16.8"W | Plum Point (40KE88) | 51°04'00" N 56°53'00" W | 3 to 4 | 2.4[c] | 47.1 | 45.4 | 32.3 |



(SR)

| | | | | | | | | |
|---|---|---|---|---|---|---|---|---|
| Eagle River (ER) | 53°33'00.0"N 56°59'13.2"W | Cartwright A (8501100) | 53°42'30" N 57°02'06" W | 3 to 4 | 0[d] | -[f] | 56.3 | 25.8 |

[a]Soil pH for the organic and mineral soil horizons determined by addition of 400 μL of aqueous 0.5 M CaCl$_2$ to a 50:50 w/w slurry of dried soil in deionised water.
[b]Environment Canada: Canadian Climate Normals, 1981 to 2010, https://climate.weather.gc.ca/climate_normals/ (last accessed: 14 July 2023).
[c]At least 20 years of measurements.
[d]The World Meteorological Organization's "3 and 5 rule" (i.e., no more than 3 consecutive and no more than 5 total missing for either temperature or precipitation).
[e]Annual precipitation averages determined using ECCC daily precipitation reports.
[f]Large quantity of missing data for this location from January 2015 to December 2016 prevents any reliable estimate.
[g]Estimated deposition rates converted to volume using DAYMET (Thornton et al., 1997, 2021, 2022).
[h]Volumes merged for 2015 and 2016 at PB and HR.

During each site visit, the slope of the sensor was confirmed to be correct, sample containers were collected and replaced with clean units, the battery and desiccant packs replaced with fully recharged devices, and the entire array confirmed operational. Four of the six sample containers (two each of OF and TF) were biologically sterilized using 1 mL of a saturated aqueous solution of mercuric chloride (HgCl$_2$) to preserve against biological growth and loss of bioavailable nutrients over the collection periods. Unsterilized sample containers (without HgCl$_2$) were used for measurements of recalcitrant species and to assess any matrix effects exerted by the preservation technique on target analyte quantitation. Collected sample volumes were measured with a 1000 ± 10 mL graduated cylinder and aliquots were collected for chemical analysis via transfer to precleaned 500- or 1000-mL HDPE containers (Nalgene; VWR International). Samples were stored at 4 °C before returning to the laboratory where they were filtered with a 1000 mL Nalgene vacuum filtration system (P/N ZA-06730-53; ThermoFisher Scientific, Waltham, MA), fitted with 0.45 μm polyethersulfone filters (PES, P/N HPWP 04700, EMD Millipore), to remove suspended solids. Filtered samples were transferred to new clean HDPE containers and stored for up to two months at 4 °C in a cold room until analysis.

## 2.4 Cleaning and Preparation of Sample Containers

All sample collection and storage containers, as well as all sample handling apparatuses, were made of HDPE or polypropylene for the quantitative analysis of target analytes. Prior to use in handling samples, these were all acid-washed in 10 % v/v HCl (P/N BDH7417-1; VWR International) followed by six sequential rinses with distilled water and ten rinses with 18.2



MΩ·cm deionised water (DIW; EMD Millipore Corporation, Billerica, MA, USA). Containers were dried by inversion on a clean benchtop protector overnight, or with protection from dust using lint-free lab wipes over container openings when necessary. Field and method blanks were collected through the addition of DIW to cleaned containers, and/or sample handling devices, in order to quantify appropriate method detection limits and to identify any sources of systematic or random contamination.

## 2.5 Measurements of pH and Conductivity

The pH and conductivity of each sample was determined using a ThermoScientific™ Orion Versa Star meter (ORIVSTAR52) interfaced with a pH electrode (Model: 8157BNUMD, Ultra pH/ATC Triode, ROSS) and 4-electrode conductivity cell (Model: 013005MD, DuraProbe, ROSS). Prior to use, the probes were calibrated daily with standard solutions specific for these probes (ThermoScientific™ Orion™ conductivity standard 1413, and pH 4, 7, and 10 buffers) and then stored between analyses according to manufacturer directions. Aliquots of 15 mL of precipitation from archived samples were subsampled into 40 mL polypropylene Falcon tubes. This was followed by immersion of a cleaned electrode for the conductivity measurement, followed by the pH probe measurement to prevent conductivity bias due to potassium chloride migration across the glass frit of the pH probe. Readings were recorded once signals had stabilized.

## 2.6 Measurements of Dissolved Organic Carbon (DOC)

Measurements of DOC were performed by catalytic combustion of samples in a platinum bead-packed quartz furnace at 720 °C to quantitatively produce $CO_2$, followed by non-dispersive infrared absorption spectrophotometry using a Shimadzu Total Organic Carbon (model: TOC-V) analyzer and an autosampler (model: ASI-V). Cleaning of materials prior to DOC determination follows the same procedure as for the sample containers. Precipitation aliquots of at least 12 mL were transferred to clean and combusted (500 °C, 5 hours) 40 mL borosilicate glass vials, then capped and stored at 4 °C until analysis. Prior to analysis, vial caps were replaced with cleaned polytetrafluoroethylene-lined septa. Inorganic dissolved carbon (e.g., $H_2CO_3$) was purged from samples by acidification to pH 2 with HPLC grade $H_3PO_4$ (20 % v/v) and bubbling with an inert carrier gas. Samples were analyzed in triplicate and quantified using calibrations spanning 0.1 to 10 or 10 to 100 ppm (mg C $L^{-1}$) with potassium hydrogen phthalate (KHP), depending on the





relative sample concentration range. Accuracy and precision were assessed using 1 and 10 ppm
KHP check standards analyzed every 10 injections, respectively. Calibrations were performed at
the beginning of every analysis day.

**3.0 Results and Discussion**

In addition to the general design advantages in the section that follows, we present the

results of various physical and chemical parameters to validate this new open source custom-built
modular system. The power consumption and snow-free performance testing are used to
demonstrate the off-grid capabilities of these samplers, as are the two-year datasets. The lower
power requirements are compared to existing commercial samplers and paired with solar top-up
to prolong the use and reduce the need to replace batteries on timescales shorter than planned
sampling duration (i.e., < 1 month). We then evaluate the automated wet deposition volumes, in
which the samplers prevent dilution during atmospheric washout events, compared to total
volumes collected from co-located samplers to depict the fractionation by volume as a function of
time. We also investigate the advantages of replicates collected across the four watersheds, using
deployments of triplicate samplers under field conditions. The ratio of collected TF to OF
replicates highlights the ability of these samplers to capture the dynamic nature of precipitation
interacting with forest canopies. Simple pH and conductivity measurements are then used as
benchmarks to situate the NL-BELT data within the established literature to emphasize the robust
operation of the samplers and impact of the selective sampling. Fluxes of DOC are then
interrogated across all four sampling sites as we demonstrate the potential of these samplers to
make measurements of more complex analyte pools that are of current interest to the atmospheric
measurement community.

**3.1 General Design Advantages**

When compared to other precipitation collection apparatuses, the automated precipitation

sampler developed in this work has several advantages. Most notable is the ability to collect
integrated samples at remote locations by exploiting its off-grid capabilities. Our approach also
maximizes the sensitivity of the rain sensor as long as electrolytes remain in the water reaching it.
The chute ensures that even if the precipitation contains ultra-trace analyte quantities, they are still
collected and quantified for an extended period when high-purity water may be deposited during



an atmospheric wash-out event. The chute does this by accumulating water-soluble materials
between rain events that require time to be completely washed off and through the release of ions
from the material itself, which ages under environmental conditions. As the conductivity of the
precipitation falls below the sensor threshold, the added ions from the chute prolong the collection
of rain past this time point. In rainfall events where extended atmospheric wash-out occurs, the
sampler lids will eventually close – preventing dilution of the sample while maintaining the
collection of analytes of interest. In application to trace pollutants, this also reduces
methodological sample preparation time as it decreases the extent to which additional handling
steps, like solid-phase extraction, are required prior to analytical determinations.
The six replicate measurements used in each array provide a means of assessing sampling
reproducibility (e.g., canopy TF has expected heterogeneity) and for multiple analyte classes to be
targeted. Various analytes, with different chemical properties and/or contamination considerations,
can be targeted by changing the materials used for the components that encounter the sample (i.e.,
lids, funnels, and sample holding containers). Replicate collection can also allow for selective
sample preservation when quantifying deposited chemical species that may be reactive, volatile,
or biologically transformed. The modularity of the overall system design also allows the collection
units to be dismantled entirely and easily reassembled on-site, minimizing logistical issues and
costs for transport to remote regions. Lastly, these collection units are cost-effective. We were able
to produce four arrays, each consisting of six collection units, at a fraction of the cost of a single
equivalent commercial off-grid automated precipitation sampling unit.
With the majority of commercial precipitation samplers requiring a source of electricity,
on-grid sample collection necessitates high infrastructure costs and/or samplers being positioned
closer than desired to point sources of anthropogenic pollution. As a result, especially in remote
locations, site selection becomes heavily restricted and expensive when factoring in all the
standard criteria, particularly with respect to the need for an easily accessible power source. Thus,
the off-grid capabilities of our samplers lends dexterity to these systems and makes deposition
sampling that follows standard siting guidelines, like those of CAPMoN or NADP but without
power, more accessible to the global atmospheric research community (Vet et al., 2014). To further
highlight and validate their capabilities, a series of fundamental performance parameters were
collected and are discussed in detail in the sections that follow.



**3.2 Power Consumption and Performance Testing**

**3.2.1 Power Consumption of Instrumental Setup**

The simplicity of the automated precipitation samplers allows for low power consumption during operation, which is particularly important for off-grid operation. The motors operating and rain sensor heating during active precipitation are the most energy-intensive elements of the system (Table 2). The integrated contribution of the motor over a month-long sampling period is however negligible compared to other components, since it is operational for short periods of 5 to 10 seconds with a current usage of only 38 mA. The continuous need to provide 5 VDC to the digital logic via step-down from 12 VDC is actually the largest power consuming component of the setup in the absence of rain. When the samplers are in the closed position, under rain-free conditions, the power consumption of the entire array is 4.66 Watts (W) and 2.86 W for transformed 115 VAC and battery 12 VDC supplies, respectively. The provision of 12 VDC to the board with a transformer for the 115 VAC application results in greater total power requirements. These values increase to 10.00 W and 5.04 W with the detection of a conductive liquid on the precipitation sensor as it heats the sensor surface to capture the active period of the event. Based on the measured power consumption, a fully charged 103 Ah AGM battery would provide at most 447 hours (or 18 days) in standby mode under rain free conditions and 294 hours (or 12 days) if the heated surface of the sensor is in continuous use (Table 2). The lower range limit is unlikely since the sensor only operates for the duration of a rain event, after which the battery is available for solar top-up again. In the fieldwork conducted here, battery life was extended through the addition of 40 W solar panels to the systems. The entire array was confirmed to be operational at the end of monthly (SR, PB, and GC) and two month (ER) integrated sampling periods on a ongoing basis, prior to exchange with a new fully-charged battery, for two years.







**Table 2.** Measured voltage, current, and power consumption of the rain sensor and circuitry in
both the idle and maximally operational state when connected to a 12 VDC battery or transformed
115 VAC. Total power demand was measured for wet and dry sensor scenarios.

|  |  |  |  |  |  |  | Total |  |  |  |
|---|---|---|---|---|---|---|---|---|---|---|
|  | **Rain Sensor** |  | **AC Outlet** |  | **DC Battery** |  | **AC Outlet** |  | **DC Battery** |  |
| **Parameters** | **Idle** | **Active** | **Idle Board** | **Motors In-Use** | **Idle Board** | **Motors in-Use** | **Dry** | **Wet** | **Dry** | **Wet** |
| **Voltage (V)** | 12 DC | 12 DC | 114 AC | 110 AC | 12 DC | 12 DC | - | - | - | - |
| **Current (A)** | 0.008 | 0.120 | 0.040 | 0.078 | 0.230 | 0.300 | - | - | - | - |
| **Power (W)** | 0.10 | 1.44 | 4.56 | 8.58 | 2.76 | 3.60 | 4.66 | 10.00 | 2.86 | 5.04 |


In comparison to two commercial samplers used by national monitoring networks, the

power requirements of our new samplers are substantially lower. The first commercial sampler we
reviewed draws a maximum of 2 A, with a ceramic heater housed within the sampler case that
draws 0.8 A constantly, resulting in an upper limit power demand of 230 W (at 115 VAC) and a
lower limit of 92 W. The commercial sampler can be upgraded to utilize a thermostated space
heater for winter operation, drawing an additional 4.2 A (480 W), resulting in a maximum power
demand of about 800 W when using a 115 VAC power supply. A second commercial precipitation
sampler reviewed is used by national monitoring networks and draws approximately 5 A, resulting
in a power requirement of 575 W at 115 VAC. The commercial and standard precipitation samplers
for deposition monitoring programs have much higher power requirements compared to those
presented in this work. The commercial samplers utilize 80 to 100 times more power. With our
lower power requirements, the new automated samplers prove to be advantageous in both on- and
off-grid sampling yet are disadvantaged in being unable to collect snow in the winter.

**3.2.2 Precipitation Sampler Performance Tests and Data Logging**

In addition to low power consumption during precipitation sampling, a supplied battery

can obtain constant power renewal when outfitted with a solar top-up that is kept exposed to
sunlight by proper orientation. At NL-BELT, adjustments were made for this during each site



visit during sample collection. During the solar top-up tests below, voltages of the sensor and
batteries were consistently monitored. Over a test period of 22 days, no appreciable decline in
battery performance of a 76 Ah unit was observed despite the detection of more than 10 rain
events during that period (Figure 3a).

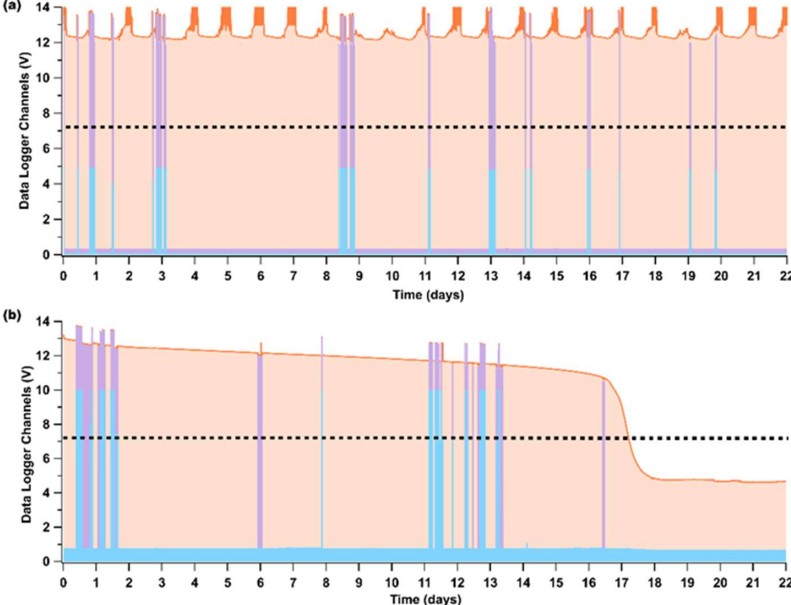


**Figure 3.** Performance of off-grid precipitation samplers during sample collections from **(a)** 13
July to 7 August 2018, using a 76 Ah battery and solar panel top-up and **(b)** 22 January to 13
February 2019 with a 103 Ah battery and no solar panel. Battery voltage (shaded orange) is
elevated above 12 VDC when charging, or decreases over time when no solar panel is used and
precipitation is sensed/collected. The 12 VDC rain sensor relay signal (purple) and the open
sampling lid switch voltage (blue) indicate active periods of detected precipitation. The black
dashed line indicates the 60% efficiency cut off, 7.2 V, at which the battery should be recharged.

In comparison, winter sampling with these devices is not recommended without substantial

investment in a sufficient power density provided high-performance cold weather batteries. The
lack of sunlight during winter at higher latitudes also negates the use of effective small scale solar
top-up. Our tests show that when the samplers were deployed without a solar backup under snow-
free winter conditions (temperatures ranging from -17.8 to 7 °C), with a 103 Ah battery, the off-
grid system only lasted for 17 days. At this point, the larger capacity battery was fully depleted by
frequent snow and rainfall – probably due to the heated precipitation sensor requiring additional



energy to phase change snow and ice to water and then to evaporate that water. This depletion
occurred despite housing the battery in an insulated enclosure during the test. In addition, on days
6 and 16, the precipitation sensor relay was activated but the lid did not rotate to the open position
(Figure 3b, blue trace). This could have been because the precipitation event was not intense
enough for the lid to open fully and trigger the 5 V lid open switch or because of snow and ice
buildup around the lids resulting in them being unable to physically open. Overall, these samplers
may be possible to deploy during the winter if line power can be supplied. Such a deployment
would further necessitate that the sampling funnel be heated to render a liquid sample for collection
in the jugs in addition to the sensor chute to prevent snow and ice accumulation. A heated funnel
would also prevent snow or ice accumulation on top of the automated lids. Together, such power
hungry requirements for winter operation exceed simple off-grid use with a battery package that
is easily transported into and out of remote field sites.

**3.3 Comparison of Sample Collection Volumes**


The automated samplers were collocated with total deposition samplers and deployed
across the experimental forests of four NL-BELT regions during the 2015 and 2016 growing
seasons to observe deposition trends. In addition, we compare these observations to the long-term
climate normals reported by ECCC and estimated deposition at 1 km x 1 km resolution from the
DAYMET reanalysis model (Table 1). Three automated samplers were deployed in the open to
collect incident precipitation (OF) and another three under the experimental forest site canopy
(TF). The mean OF volumes of triplicate measurements from south to north were 1.42, 1.38, 1.31,
and 0.79 L, whereas the corresponding TF volumes were generally similar in magnitude at 0.96,
0.98, 1.02, and 1.13 L, for the 2015-16 sampling period (Figure 4). It is evident that the volume of
precipitation decreased as latitude increased for OF samples, whereas the opposite relationship
was observed in TF samplers, although the absolute volumes are more comparable in magnitude.
The total deposition volumes collected were as expected, decreasing from south to north in
agreement with the expectations from the long-term normals and comparable to the estimates from
the DAYMET model (Table 1), where the largest integrated volume of precipitation was collected
in the lowest latitude (GC) and a lower amount in the highest latitude (ER), with the intermediate
sites (HR and PB) having the lowest inputs overall during this observation period. Total annual
deposition volumes collected by our deployed samplers from south to north in 2015 were 39.5,



39.4, 31.9, and 17.5 L, while in 2016, they were 51.7, 37.8, 32.8, and 34.2 L. Total deposition
volume collected from HR was used for comparison to automated sample volumes collected at PB
in 2015, as they both share the same watershed. This approach had to be taken, as the HR site was
initially planned for full experimental use before becoming inaccessible in early 2015. The relative
error between the two sites for samples collected in 2016 was ±15% (24.6 L in PB and 32.2 L in
HR), comparable to the reproducibility we observe for replicates collected within a given site (see
below). The total deposition samplers were installed in HR in late 2014 and the automated samplers
were then set up at PB. Despite this, there is good agreement between the trends in predicted
deposition values by DAYMET with the measured values, although the absolute amounts from
these are systematically lower in all of our observations. Sufficiently continuous measurements
from ECCC stations nearby each site are challenging to obtain for the 2015-16 period. When
available with greater than 80% coverage, the ECCC datasets both agree and disagree with our
observations in GC and SR, respectively, suggesting that there is substantial deposition volume
heterogeneity at the scale of ~10 km in this region. In SR, the disagreement with our measurements
is identical to the DAYMET model which uses ECCC observations as input data, while at GC the
ECCC measurements are identical to ours (Table S3).  Further, the discrepancy in the PB or ER
average annual precipitation volume between ECCC and those of this work and DAYMET are not
possible to interrogate due to the large quantity of data missing from the ECCC monitoring station
(35.22% in ER and 39.65% in HR/PB; Table S3). The DAYMET observations are representative
of a larger spatial scale, where our discrete samplers could be subject to heterogeneity in deposition
(e.g., orographic precipitation, driven by topography like steep slopes) or impacted by
meteorological conditions not captured by the model (e.g., undercatch driven by local winds). The
temporally-resolved volume comparisons at sampling interval timescales better-demonstrates
comparability, despite the systematic differences. The month-to-month relationships between
DAYMET (and ECCC) and our observations all showed strong correlations at all sites, with linear
regressions having $R^2$ of 0.72 at ER, and 0.99, 0.99, and 0.86 (N/A, 0.941, N/A, and 0.934,
respectively) when progressing through the more southerly sites (Table S3). The discrepancy
between DAYMET, ECCC, and our observations for total deposition were highest in the most
northerly site, where the experimental site was located on a steep slope, with only 43 % of the
predicted volume collected. At all of the sample collection sites on the island of Newfoundland, a
consistent difference was observed with 65 ± 4 % of the estimated volume collected, except at GC



where our measurements and those from ECCC are identical and starkly contrasting to DAYMET.
Overall, parsing these comparisons is difficult and demonstrates that there may be up to 55%
additional uncertainty in deposited species, should given measurements of a species be scaled for
a watershed like ER by concentration in total deposition samples. We propose, that by isolating
only the deposited analytes and using analyte fluxes instead of concentrations in precipitation
samples, that uncertainty issues in representing volumes, improves overall deposition budget
certainties. Regardless, by following the recommended siting criteria from the NADP and
CAPMoN as best as possible, the very strong agreement of our temporal trends at both annual and
monthly timescales with both comparators demonstrates the suitability of the total deposition
samplers and therefore the automated samplers for use in quantifying deposited chemical species
of atmospheric interest into the experimental sites.

The wet deposition volumes collected for the snow free period using the automated

precipitation samplers did not follow the trends in total deposition (Figure 4), as might be expected.
For the 2015 collection period from June through October, the summed volumes of OF
precipitation, from south to north across the NL-BELT, were 25.4, 10.9, 20.4, and 2.2 L, while in
2016 they were 17.3, 30.4, 13.5, and 5.1 L. There are three reasons as to why the measured wet
deposition volumes do not follow the total deposition trend across the transect. First, these
samplers are designed specifically to collect only conductive precipitation (i.e., containing
conductive atmospheric compounds) not total/bulk precipitation. As a result, the OF wet
deposition volume collected across the sites is mostly below 50% of total volumes collected, while
TF volumes are similar in magnitude or lower than that of OF (Figure 4). The wet deposition
fraction collected was variable within and between regions, sometimes less than 10%, despite large
volumes collected in total and presumably due to intense atmospheric washout that this region is
well-known for. Second, the NL-BELT total deposition trend estimated using the ECCC long-term
climate normals represents a 30-year period (Bowering et al., 2022) while the automated volume
measurements here represent two years of targeted conductive precipitation collection. The
combined summed volumes of targeted conductive wet deposition across the 2015 and 2016 field
seasons were 42.7, 41.3, 33.9, and 7.3 L, somewhat better reflect the expected precipitation trends
within the transect (Table 1). Lastly, our monthly automated wet deposition sample collection
periods occurred from June through November and so it is temporally incomplete with respect to
the substantial amount of precipitation volume deposited as snow delivered during the winter



(Table S3). The discrepancies between the long-term trends and our shorter-term observations
therefore make sense as they are sensitive to interannual changes in synoptic scale transport and
rainwater solute loadings, as exemplified by the volumes collected in SR in 2015 (Figure 4b) and
PB in 2016 (Figure 4c). Overall, for the automated sampler observations on a per-year basis, there
is no consistent trend between site latitude and the volume collected in either OF or TF. This is
unsurprising as they are dependent on the conditions that drive the rate of atmospheric wash-out
and presence of conductive solutes.

The automated OF wet deposition volumes collected each year have peak values that range

from 1 to 4 L with an overall variability of $\pm$ 33% for any triplicate of samples across the entire
dataset. Wind is known to generate bias in gauge-based precipitation measurements where
unshielded precipitation gauges can catch less than half of the amount of a shielded gauge (Colli
et al., 2016). A windscreen design for obtaining rainfall rates – and thus, volumes – to be more
reproducible could be considered in future deployments of our developed samplers, similar to
recently reported innovations for smaller rainfall rate devices (Kochendorfer et al., 2023). This
would however increase costs and logistical considerations in deploying the developed devices,
which currently operate synonymously to deposition systems employed by government monitoring
programs. In addition, collection of replicate samples allows our observations to span a wider
physical area, reducing the impact of confounding variables such as wind speed in comparison to
a more typical sample size of one for many field collections. Imperfect siting and lack of shielding
is necessary where remote field sampling prevents the setup of such infrastructure. Across our 33
sample collection periods, our replicate relative standard deviations (RSDs) follow a log-normal
distribution where volume reproducibility is typically within $\pm$ 12.5% and almost always within $\pm$
31.5% (Figure S11). A few outliers with higher variability skew the overall view of volume
precision. Out of 33 OF samples collected, 8 have RSDs greater than 40.5% and 2 of those 8 have
RSDs greater than 100%. As a result, the deployment of triplicate samplers provides researchers
with a better opportunity to implement quality control as they can reduce bias in the event of
dynamic OF. While the effect of wind is reduced, additional factors can drive variability when the
samplers are placed under a forest canopy for TF collection.

To demonstrate canopy dynamics impacting interception volumes within the sampling

sites, the ratio of throughfall to open fall (TF/OF) volume was compared amongst our total pool
of 31 samples. This group of samples encompassed the monthly average TF/OF values for each



set of triplicate samplers, at all four sites, from 2015 to 2016. These measurements were then split
into two separate populations – samples that have a TF/OF less than one (n=24) and those that
have a TF/OF greater than one (n=7). The samplers were positioned identically between years and
no single sampler was reproducibly found in the second population. In the first population, the
fraction collected was $56 \pm 21\%$ (ranging from 19 to 88%), likely due to the known processes of
canopy and stem interception (Eaton et al., 1973; Howard et al., 2022). For example, in two young
balsam fir-white birch mixed forest stands, the amount of precipitation intercepted by the forest
canopy, in similar snow-free conditions, was estimated to be $11 \pm 5\%$ (Hadiwijaya et al., 2021).
In mature boreal forests, 9% to 55% of rainfall can be intercepted by the canopy (Pomeroy et al.,
1999). Relevant to deposition of atmospheric constituents, Pomeroy et al. (1999) also reported that
up to 70% of intercepted rainfall may evaporate directly from the canopy, which can leave behind
non-volatile rainfall solutes. Wet deposition that undergoes stemflow (SF) proceeds down the
branches, stems, and/or trunks of a plant, transferring precipitation and nutrients from the canopy
to the soil at the trunk or stem base (Ciruzzi and Loheide, 2021). These known mechanisms of
canopy interception ultimately reduce the amount of precipitation reaching the ground as TF, and
thus, the explain the smaller volumes found in our samplers compared to the OF measured
simultaneously. In contrast, the fractions that ranged from 108% to 424%, averaging 186%,
demonstrates a different aspect of the highly dynamic nature of canopies where they can sometimes
intercept rainfall like an impermeable surface to act as a funnel, guiding large volumes of
precipitation on to the ground, or in this case into the TF samplers (Metzger et al., 2019).

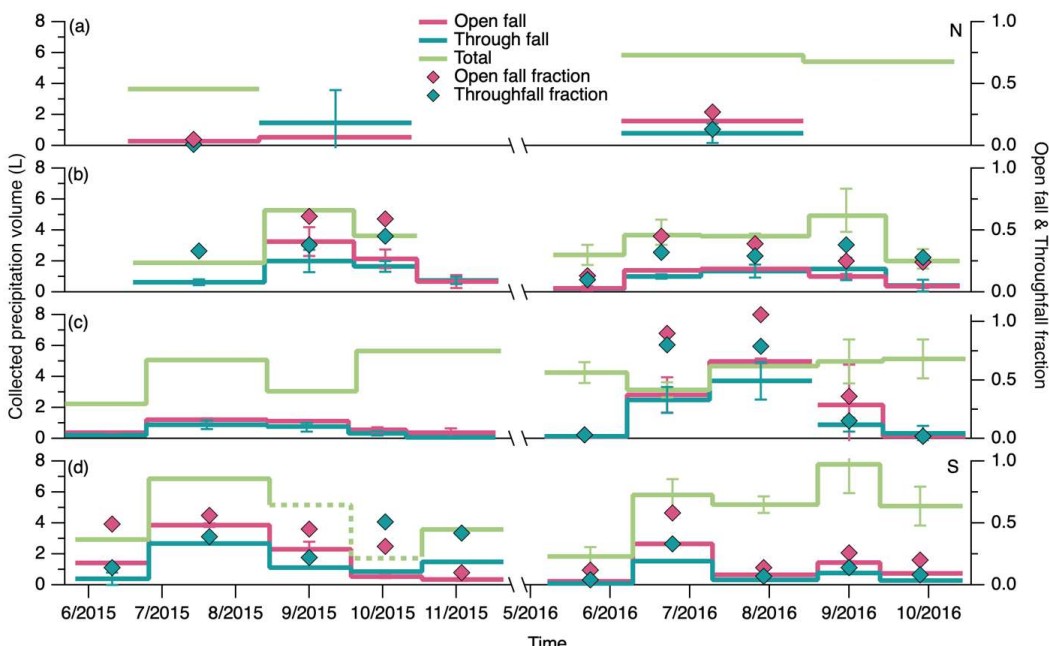

**Figure 4.** Average volume collected from replicate automated samplers deployed from June 2015 to October 2016, from north (N) to south (S), at the NL-BELT field sites: **(a)** ER, **(b)** SR, **(c)** PB, and **(d)** GC. The red trace represents open fall, teal for throughfall, and light green for total deposition (the sum of conductive and non-conductive precipitation). The total precipitation volume depicted for PB, from July 2015 to November 2015, were collected at the nearby HR site in the same watershed since no total deposition measurements were in place at PB during this period. The missing volume for GC in 2015 was estimated from the determined ECCC station linear relationship and is presented as a broken line. The fraction of precipitation collected as open fall or throughfall, compared to the total deposition (right axis), are represented by diamonds of the corresponding color. Error bars represent the standard deviation of three measurements from replicate samples. The axis break spans the winter months when the off-grid automated samplers were stored.

## 3.4 Characterizing Chemical Parameters from NL-BELT

In addition to assessing physical parameters, chemical parameters were also evaluated in this work. Conductivity and pH are measurements commonly made on precipitation samples collected from the field and so incorporating them into our analysis is useful for instrumental validation. Additionally, with increasing recognition of their importance as a proxy for ROC estimation, and in biogeochemical carbon budget closure, DOC flux measurements were used to compare against a limited number of prior reports, each using different sampling or data interpretation strategies. These chemical measurements were also made in an underrepresented





part of the world in terms of atmospheric deposition sampling and are useful additions to the
overarching study of precipitation chemistry.

### 3.4.1 Precipitation pH

The deposition of atmospherically persistent pollutants and biogeochemically relevant species
to the Earth's surface, or even $NO_3^-$ and $SO_4^{2-}$ historically, can affect the environmental health of
soil, air, and water. With the pH range of natural rainwater in equilibrium with atmospheric $CO_2$
expected to be between 5.0 to 5.6, acid rain is defined by values lower than this (Han et al., 2019).
Traditionally, the extent of acidity depended on the intercepted atmospheric concentrations of
$HNO_3$ and $H_2SO_4$. In any case, monitoring acidity and deposition is especially relevant in remote
regions, where major uncertainties and gaps in deposition measurements and global ion
concentrations exist (Escarré et al., 1999; Vet et al., 2014).  A change in pH can modify the
chemical state of many pollutants, altering their transport, bioavailability, and solubility (Guinotte
and Fabry, 2008). For example, this can increase exposure and toxicity of metals and nutrients in
marine habitats which can go undetected for longer periods in remote areas.
Most TF samples were observed to have slightly higher pH (4.74 to 5.99) than those from OF
which had pH values ranging from 4.14 to 5.71 (Figure 5).  The four remote NL-BELT sites are
dominated by balsam fir trees underlain by humo-ferric podzol soil with pH ranging between 3.0
and 4.5 (Table 1). In comparing the pH of podzolic soil to that of the collected TF, it was observed
that on average, the collected precipitation has a slightly more basic pH – with rare exceptions
(e.g., July and September 2015 PB pH of 3.69 and 4.26, respectively, and the July 2015 GC with
pH of 4.12). Excluding these exceptions, the TF precipitation pH ranged from 4.74 to 5.99 with
no major variations observed spatially between the four sites, or temporally between seasons or
years (Figure 5). The pH values reported at each of the NL-BELT field sites are comparable to
recent OF measurements made at CAPMoN sites in Nova Scotia and Newfoundland and Labrador,
where the reported pH of precipitation ranged from 4.44 to 5.19 (Houle et al., 2022).
Precipitation components have three possible fates upon entering soil, such that: i) acidic
components can be neutralized by free bases such as calcium carbonate ($CaCO_3$); ii) they can pass
into ground water; or iii) undergo exchange reactions with compounds already present on soil
surfaces. In particular, exchange of cations on negatively charged clay particles can occur and
impact soil properties, when exposed to acidic rainwater. This soil property is typically termed the



cation exchange capacity (CEC) and can impact the stabilization of organic matter (McFee et al.,
1977). There could be interactions between soil properties, like CEC, and precipitation
components, although this depends on the composition of the chemical system. At NL-BELT, the
CEC is lowest in the north at ER (7.8 cmol kg$^{-1}$) and highest in the next most northerly site at SR
(19.2 cmol kg$^{-1}$) and could be related to soil organic matter composition as well as the parent
material from which the mineral soil is derived (Patrick et al., 2022), in addition to its history
interacting with the ions delivered by incident rainfall. Upon interacting with precipitation, if the
water is acidic from the dissolution of strong acids like $HNO_3$ and $H_2SO_4$ to yield $H_3O^+$,
undesirable cations such as $Al^{3+}$ could be liberated from mineral soils, in addition to $H_3O^+$
displacing beneficial nutrient cations for plants at exchanges sites throughout the soil, such as $Mg^{2+}$
and $Ca^{2+}$. However, in marine environments such as this, substantial amounts of dissolved cations
are also deposited (Feng et al., 2021) which act as spectator ions in aqueous solutions, having no
bearing on the measured acidity of precipitation. As a result, it is challenging to infer the extent to
which incident precipitation influences soil CEC at these sites without further experimental study.

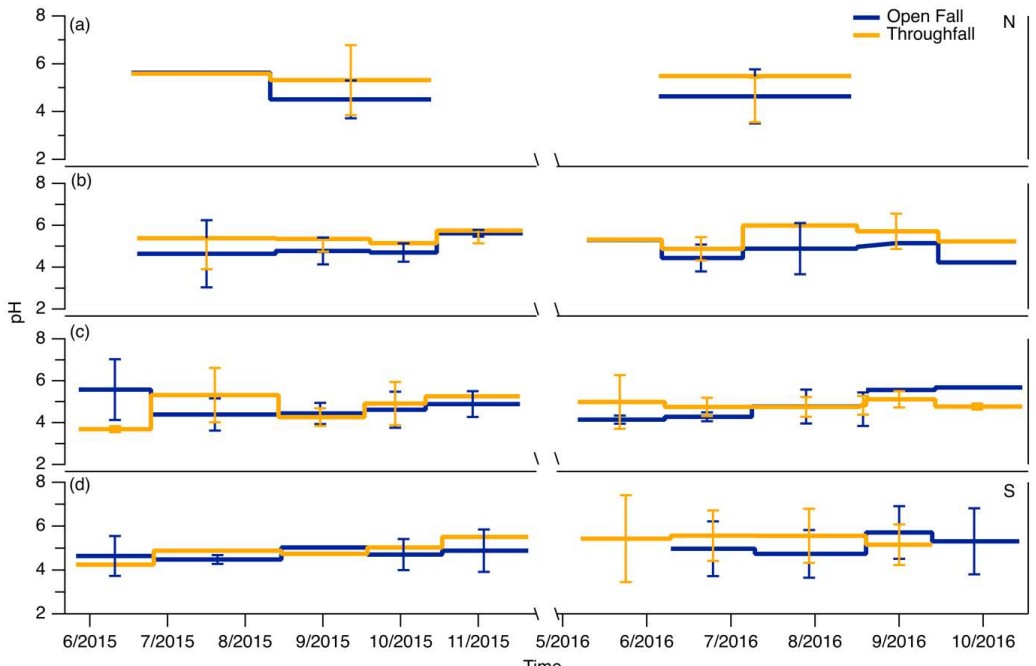

**Figure 5.** Average pH values from replicate samples collected at each NL-BELT field site, from
north (N) to south (S), at **(a)** ER, **(b)** SR, **(c)** PB, and **(d)** GC, from June 2015 to August 2016.



Open fall collections are represented using the solid blue trace whereas the orange trace is the pH
of the precipitation collected as throughfall under the balsam fir canopy.

The more basic TF overall is expected, as it has been found that up to 90% of $H_3O^+$ in
precipitation can be absorbed by leaves while passing through the canopy (Cappellato et al., 1993).
Foliar leaching, the release of ions from leaves, has been commonly reported for base cations such
as $Mg^{2+}$, $K^+$, and $Ca^{2+}$ while being minimally observed for other ions such as $Cl^-$ and $SO_4^{2-}$ (Carlson
et al., 2003). Mechanisms for foliar leaching include passive cation exchange of $H_3O^+$ with, for
example, cells in the interior of the leaf (Burkhardt and Drechsel, 1997). Additionally, alkaline
dust – deposited on the leaves of the canopy, can decrease the acidity of TF precipitation. Such
dust can accumulate on leaf surfaces as a result of anthropogenic (i.e., industrial processes) or
natural (i.e., wind erosion) sources (Csavina et al., 2012), so that precipitation passing through the
canopy can interact with it (e.g., $CaCO_3$); thus, neutralizing acidic species and increasing the TF
pH observed in our automated samplers.
The pH of the collected precipitation appears to be similar in both TF and OF as a function of
time – despite the potential for foliar leaching and dust dissolution in the canopy. The same
chemical components may be setting the pH, as these measurements do not vary much seasonally,
geographically, or temporally. As pH is a long-studied measurement, its purpose in this work was
to validate the sample quality from our described collection approach, rather than drive any
scientific objective. Nevertheless, while the NL-BELT measurements demonstrate a recovery
compared to rainwater pH in 1980s eastern North America – prior to $NO_x$ and $SO_2$ regulation (pH
from 4.1 to 5.0; Barrie and Hales, 1984), the present-day pH remains lower than expected for
natural rainwater (~5.6; Boyd, 2020). Keeping in mind the successful environmental policies
limiting $SO_2$ and $NO_x$, leading to considerable decreases in atmospheric concentrations of $H_2SO_4$
and $HNO_3$, a modern view on the trajectory of continental U.S. cloud water composition and pH
has recently been reported (Lawrence et al., 2023). Across the U.S. and eastern Canada,
measurements of anion molar charge equivalents have been lower than cations – a potential
explanation being an increase in the presence of weak organic acids which commonly have pKa
values near 4 (Feng et al., 2021), an outcome we have also observed in aerosol sample chemical
composition from Atlantic Canada (Di Lorenzo et al., 2018). With the frequency of acid rain
having a pH < 5 decreasing over the past 20 years, these recently reported measurements depict
deposition composition shifting away from a 'linear' chemical regime dominated by $H_3O^+$ and



$SO_4^{2-}$ towards a 'non-linear' regime designated by low acidity, moderate to high conductivity,
potentially weak acid-base buffer systems, and increasing base cation and TOC concentrations
(Lawrence et al., 2023). It would seem the evolving chemical contributors to global rainwater pH
remain an open line of investigation.

### 3.4.2 Precipitation Conductivity

In all the collected OF and TF precipitation samples, across all four NL-BELT sites, the
average measured conductivity values ranged from 21 to 166 uS/cm (Figure 6). With the typical
conductivity of surface and drinking waters being between 1 to 1000 µS/cm (Lin et al., 2017), and
typically below 200 uS/cm in stream water measurements within the watersheds of each of the
NL-BELT sites, our observations are comparable and fall within the expected range. Our field
blanks – encompassing a variety of materials and apparatuses, and our cleaning procedures,
routinely produced conductivities of $9 \pm 5$ µS/cm. The conductivity of saturated $HgCl_2$ in water
(at 0.1% vol/vol) was $13.6 \pm 0.4$ uS/cm, which is also comparable to our field blanks and less than
what was observed for our samples. Even with this background correction applied, the conductivity
values presented here are expected to be similar to or higher than what would typically be found
in rainwater (4 to 150 µS/cm; Beverland et al., 1997) as the rain sensor deliberately selects for
precipitation containing ionic chemical components with conductivity greater than 1.0 uS/cm,
while excluding pure water during atmospheric washout, which would dilute the dissolved solutes
in the wet deposition sample and lower the resulting conductivity values. The overall
comparability between our range and those previously reported, where the lower limit is slightly
higher in our dataset, demonstrates that the principle of operation of our instrument is robust. It
decisively collects precipitation with the property of conductance indicating dissolved ionic solutes
of interest to atmospheric chemical processes.

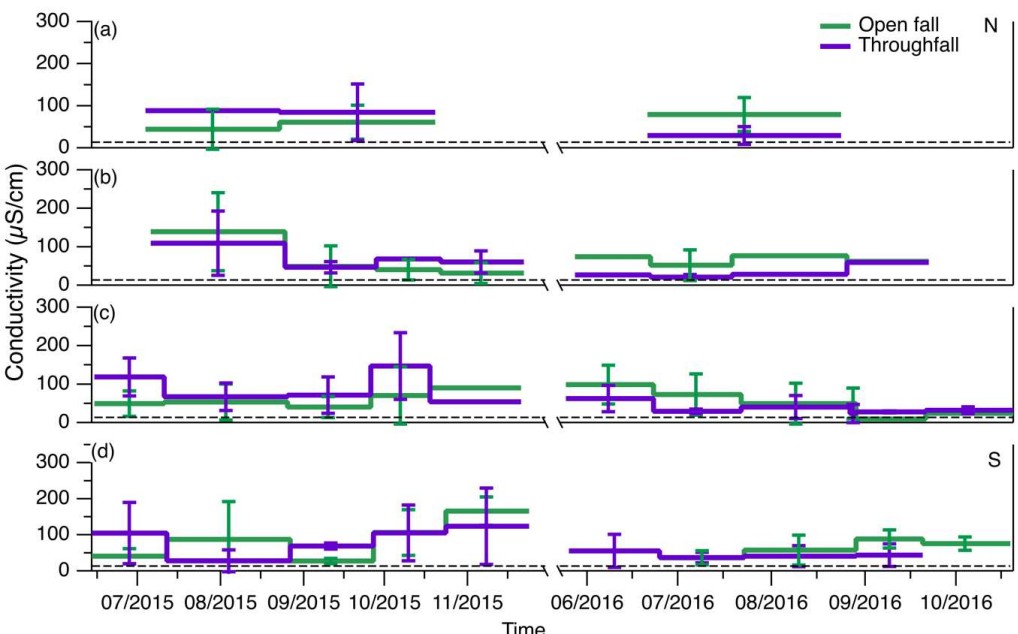

**Figure 6.** Average conductivity measured from replicate automated samplers at the NL-BELT
field sites, from north (N) to south (S), at **(a)** ER, **(b)** SR, **(c)** PB, and **(d)** GC, from June 2015 to
October 2016. The green trace represents open fall samplers whereas the purple trace represents
throughfall samples. The error bar represents the standard deviation between replicate
measurements. The dashed black line represents the upper threshold of conductivity (13.6 µS/cm)
that arises due when an addition of saturated aqueous $HgCl_2$ is made to microbially sterilize
samples. Note that all samples have conductivities equivalent to or higher than 13.6 µS/cm.

**3.4.3. Wet Deposition of Dissolved Organic Carbon (DOC) at NL-BELT**

The concentration of DOC in OF and TF precipitation, across all four sites, ranged from 3

to 46 mg L$^{-1}$ and 5 to 65 mg L$^{-1}$ with averages of 16 ± 10 mg L$^{-1}$ and 22 ± 12 mg L$^{-1}$, respectively
(Table 3). Concentrations are influenced by the volume collected and are not useful when
discerning deposition trends and/or mechanisms. The concentrations were converted to elemental
fluxes (mg C m$^{-2}$ d$^{-1}$) using the volume of precipitation collected, the area of the funnel and the
number of sampling days of each sampling period (Figure 7). The total flux for each sample period
was summed to determine the annual flux and ranged from 600 to 4200 mg C m$^{-2}$ a$^{-1}$ across the
study sites for the snow free period (Table S4).

The TF DOC fluxes were enhanced compared to the corresponding OF samples as

precipitation was intercepted by the forest canopy, with fluxes higher in TF samples by 600, 400,
and 400 mg C m$^{-2}$ a$^{-1}$ at GC, SR, and ER, respectively (Table S5). The accumulation of water-





soluble organics on forest canopies that increases DOC detected in TF could originate in part from organic carbon-containing compounds aged through oxidation reactions in the atmosphere, which increases their water solubility and propensity for surface interactions. In periods without substantial rain, these oxidized organics deposit effectively to the high surface area of forest canopies, contributing to the elevated DOC measured in TF. Additionally, non-volatile organics left behind from evaporated precipitation intercepted by the canopy could also contribute. Conversely, other mechanisms within the forest could result in enhanced DOC in TF. Recently, Cha et al. (2023) utilized a mass balance approach to determine whether DOC deposition is driven by canopy leaching (i.e., soluble tree resin, leaf exudates, internal tissues and microbes) or dissolution of dry deposited gases and $PM_{2.5}$ on plant foliage into rainwater. It was found that canopy leaching is the major contributor to TF DOC, accounting for ~83% of throughfall DOC. Whereas, $PM_{2.5}$ and rainwater only accounted for ~3 and 14%, respectively, while dry deposited gases were not considered. This suggest that internal cycling of DOC within the forest could an important source of DOC to the throughfall soil interface (Cha et al., 2023). It is possible that a similar mechanism may be responsible for the elevated levels of DOC in TF at the NL-BELT sites, but we cannot explicitly distinguish between internal cycling versus external deposition in the current study.

A notable exception was observed at PB, where the DOC fluxes in the open fall sample was enhanced up to 1800 mg C $m^{-2} a^{-1}$ when compared to the TF in 2016. This may be attributed to a difference in forest type within this NL-BELT region being black spruce (*Picea mariana*) instead of balsm fir (Bowering et al., 2023). Some studies have suggested that forest type could be a major factor affecting DOC variability (Arisci et al., 2012; Sleutel et al., 2009). Specific differences in canopy height, leaf area index, canopy structure and the shape of leaves and needles could drive DOC differences between forest types (Smith, 1981; Erisman and Draaijers, 2003; Sleutel et al., 2009). The elevated levels in OF samples relative to TF within PB are consistent with idea of uptake and/or leaching of canopy DOC in the internal cycling of DOC, while the enhanced TF at the rest of the sites is more difficult to observational constrain the participating processes.

Episodic events, such as polluted air masses from wildfires could also result in elevated deposition of DOC. It is estimated that ~116 – 385 Tg C $a^{-1}$ is produced globally due to the incomplete combustion of biomass during landscape fires (Santín et al., 2016; Coward et al., 2022).





Several studies have associated enhanced DOC levels with wildfires (Gao et al., 2003; Moore,
2003; Wonaschütz et al., 2011; Myers-Pigg et al., 2015). More recently, Coward et al. (2022)
measured DOC in Pacific surface waters along the California coastline and observed 100 to 400
% increases in DOC concentration, when compared to pre-wildfire conditions. It is possible that a
similar biomass burning plume that underwent atmospheric washout, could be responsible for the
enhancement in the observed DOC at NL-BELT, overlaid on a background more typical of
seasonal oxidation of biogenic DOC. This also coincides with the seasonal variability observed in
OF samples from the same summer where elevated levels of DOC were measured. For instance,
the DOC deposition at PB for August 2016 was 4800 mg C m$^{-2}$, whereas the total deposition for
the same year was 7800 mg C m$^{-2}$ a$^{-1}$. This single period accounts for 62% of the total DOC
deposition at this site. This underscores the pivotal role that episodic transport may play in
influencing the dynamics of DOC deposition, particularly with a warming future where wildfires
are more prevalent.

The deposition trend observed in the current study also highlights the complexity of the

varied drivers of atmospheric ROC, in which some months have more DOC in TF versus OF and
occasionally the opposite is observed. Generally, we observed similar fluxes in both samples –
suggesting that the amount of deposited carbon is comparable. Although the volume of
precipitation captured in TF samplers are generally lower when compared to the corresponding
OF samplers, the deposition flux of DOC is greater in TF samplers. With DOC enhanced in TF
samples, the values reported here could be an underestimation of the amount of carbon reaching
the forest floor during precipitation events due to competing processes within the canopy. One
such process is stemflow (SF), where a fraction of precipitation intercepted by the forest canopy
is funneled over the bark of the tree surface to the base of the tree stem (Oka et al., 2021). Although,
SF was not measured in the current study, several studies have demonstrated that DOC
concentrations are enhanced in SF when compared to the corresponding TF and bulk precipitation
samples (Stubbins et al., 2017; Van Stan and Stubbins, 2018; Ryan et al., 2021). Additionally, we
cannot rule out that the chemical speciation differs between OF, TF, and SF even if the DOC values
are similar, but such insights require more selective instrumentation for chemical analysis (e.g.,
high resolution mass spectrometry).

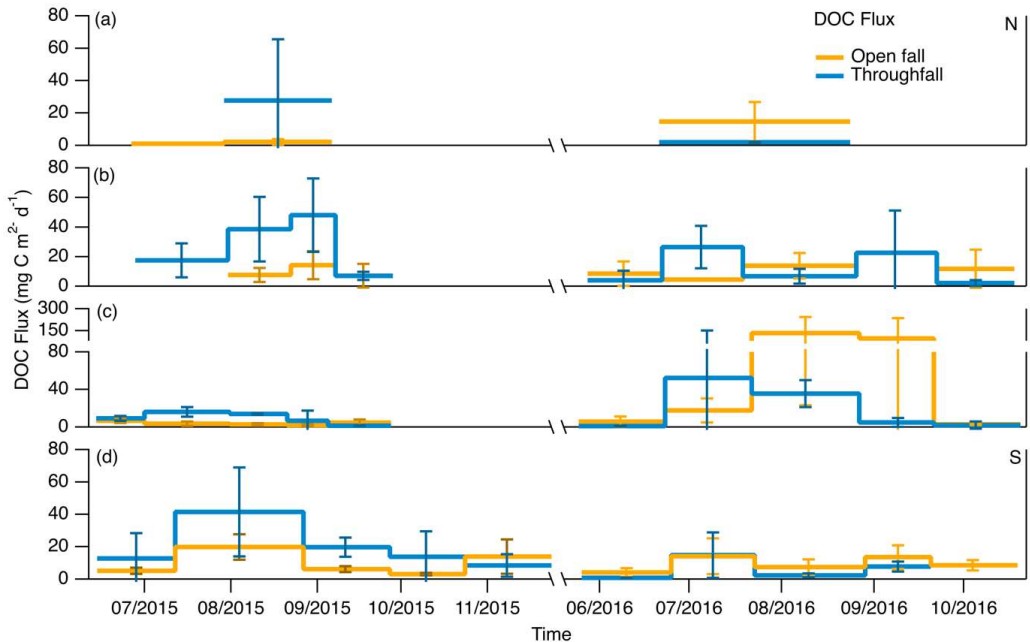


**Figure 7.** Average DOC fluxes (mg m⁻² d⁻¹) in replicate samples collected at the NL-BELT field
sites, from north (N) to south (S), at **(a)** ER, **(b)** SR, **(c)** PB, and **(d)** GC, from June 2015 to August
2016. The yellow trace represents samplers that were placed in the open without any obstruction
whereas the blue trace represents samplers that were placed under the canopy. Error bars represent
the standard deviation of three measurements from three independent samples.

Mean annual DOC fluxes were generally similar to those reported in some other boreal

forests (Table 3). These include Finland, with work in stands that consisted mainly of Scots pine

(*Pinus sylvestris L.*) (mean OF 2.32; TF, 4.35 g C m⁻² a⁻¹; Pumpanen et al., 2014), as well as in

Mont St. Hilarie, Québec (mean OF 0.49; TF 2.05 g C m⁻² a⁻¹; Dalva and Moore, 1991), which also

consisted of a variety of tree species such as yellow birch (*Betula allenghanien*), red maple (*Acer

rubrum*), and sugar maple (*Acer saccharum*). Conversely, the annual fluxes were orders of

magnitude lower than measurements made at the University of Georgia (23 to 48 g C m⁻² a⁻¹) which

has a subtropic climate consisting mainly of southern live oak (*Quercus virginiana Mill.*) and

eastern red cedar *(Juniperus virginiana L.)* occasionally hosting dense epiphytes (Van Stan et al.,

2017). This highlights the potential variability to expect when measuring DOC in different forest

systems, as the annual DOC fluxes vary depending on factors such as climate, tree species

composition, and environmental conditions.

These results underscore the pivotal role the off grid custom-built automated deposition
samplers can play in advancing scientific research, particularly in precipitation monitoring and
analysis. The automated system enabled long term continuous sample collection in remote
locations, which was previously challenging to attain due to the need for frequent human
intervention and resources required to regularly access these experimental forest stands. These
samplers also allowed us to compare DOC through replicate measurements in TF and OF samples
which sheds light on the potentially different DOC deposition chemistries within the NL-BELT
region. The automated system better maintains the integrity of DOC in the samples, since the
introduction of forest litter could result in a positive bias for DOC in the collected precipitation.
The use of replicates also results in more robust scientific conclusions and broader applicability of
the results, and they can be obtained for a fraction of the cost of a commercial equivalent,
highlighting the contribution these automated systems are capable of when applied to current
precipitation monitoring. As a result, these samplers show promise in the quantification of
biogeochemical and anthropogenic chemical species of interest, which will be visited in future
manuscripts drawing from the samples presented in this dataset, and others since obtained, but are
beyond the scope of this manuscript in demonstrating the performance of this new instrumentation.

**Table 3.** Concentrations (mg C L$^{-1}$) and annual fluxes (g C m$^{-2}$ a$^{-1}$) of DOC in precipitation (P),
open fall (OF), throughfall (TF), and stemflow (SF) collected at forested sites. Where volumes are
not available for other studies, fluxes are not possible to calculate. The values reported in the
current study are the estimated DOC flux for the wed deposition sampling period (~June through



October) for each year and therefore represents the lower limit of DOC deposition, as the dataset
excludes snow.

| Site | Type | Mean Concentration (mg C L$^{-1}$) | Annual Flux (g C m$^{-2}$ a$^{-1}$) | References |
|---|---|---|---|---|
| Grand Codroy, NL, Canada (2015 to 2016) | OF | 12.83 | 1.56 | This study |
| | TF | 23.40 | 2.20 | |
| Pynn's Brook, NL, Canada (2015 to 2016) | OF | 19.98 | 4.21 | |
| | TF | 21.24 | 2.44 | |
| Salmon River, NL, Canada (2015 to 2016) | OF | 16.14 | 1.33 | |
| | TF | 21.00 | 2.65 | |
| Eagle River, NL, Canada (2015 to 2016) | OF | 11.59 | 0.53 | |
| | TF | 28.26 | 0.86 | |
| Mont St. Hilaire, QC, Canada (1987) | P | 2.00 | 0.49 | Dalva and Moore, 1991 |
| | TF | 12.13 | 2.05 | |
| | SF | 40.10 | 0.10 | |
| Northern China (2007 to 2008) | P | 2.4 to 3.9 | 1.4 to 2.7 | Pan et al., 2010 |
| Coulissenhieb, Northeast Bavaria (1995 to 1997) | P | 2.70 | - | Michalzik and Matzner, 1999 |
| | TF | 15.20 | - | |
| Hobcaw Barony, South Carolina, USA (2014 to 2015) | P | 1.20 | - | Chen et al., 2019 |
| | Pine TF | 26.00 | - | |
| | Oak TF | 38.8 | - | |
| University of Georgia, USA 2015 to 2016 | TF Epiphyte Oak | 17 | 23** | Van Stan et al., 2017 |
| | TF Bare Cedar | 20 | 32** | |
| | TF Epiphyte Cedar | 54 | 48** | |
| SMEARII Site, Southern Finland (1998 to 2012) | P | 3.24 | 2.32 | Pumpanen et al., 2014 |
| | TF | 10.10 | 4.35 | |

** Estimated DOC yield for 2016 (g C m$^{-2}$ a$^{-1}$) where sampled storms values (g C event$^{-1}$) were scaled
to an annual deposition value using meteorological data and a linear rainfall-DOC yield relationship.

**4 Conclusions**

This paper presents a cost-effective automated deposition sampler for continuous

collection of precipitation. An open-source procedure and schematics for building these samplers
is provided alongside the rationale for selecting the materials in the current study to target analytes
of scientific interest in wet deposition samples. These low-power systems are demonstrated in
being capable of continuous off-grid use for sample collection over two years at the NL-BELT
experimental sites, with replacement of battery power packs monthly or bimonthly, with on-grid





performance also provided for comparison. The resulting systems enhance the accessibility of
automated wet deposition samplers to scientists globally and this work highlights their robust
performance in collecting and preserving rainwater conductivity and pH, alongside providing
measurements of DOC from this understudied region that builds a broader picture of the
atmosphere-surface exchange of this biogeochemical pool across the NL-BELT. Comparability
and complementariness of our results to well-established and current measurements of interest like
DOC, demonstrate their efficacy. The samples collected in this work from this new instrumentation
are expected to be used further in several complementary and novel environmental monitoring
studies in the future to extend our biogeochemical analysis, but also to study the transport of other
anthropogenic pollutants of emerging interest, which are beyond the scope of describing our new
platform. For the broader deposition-motivated community, the instrument design also allows for
easy cost-effective modification of the number of replicate samplers, the material composition of
all surfaces the aqueous samples interact with, as well as preservation strategies, depending on the
analyte of interest. The capacity to autonomously collect wet deposition, in addition to traditional
bulk deposition samples can shed light on competing wet and dry deposition processes. Should
on-grid capacity suit scientific objectives, these samplers are anticipated to be possible for use
year-round when paired with more power-intensive strategies to facilitate solid to liquid phase
transfer for detected and collected precipitation in the winter.



*Data availability.* The data are available from the corresponding author (TV) on request.

*Author contributions.* AC, DP, and ML performed the data analysis. AC and DP wrote the
manuscript with contributions from all authors. Sampler design and construction were led by TV,
with assistance from BP and RH for initial prototypes, DP and ML for the revised iteration, and
AC for the final modular control boards. Sample collection and associated characterization
measurements were performed by BP and TV. Conceptualization and conduct of the sampling
experiments were made by TV, CY, KE, and SZ. All authors were involved in examining and
reviewing the results. All authors were involved in editing the paper.

*Competing interests.* The contact author has declared that none of the authors has any competing
interests.

**5 Acknowledgements**

Funding for this work was provided by the Newfoundland and Labrador Department of
Agrifoods and Forestry, Centre for Forestry Science and Innovation (Project 221269), and the
Harris Centre at Memorial University. T. C. VandenBoer was supported for this work in-part
through a Government of Canada Banting Postdoctoral Fellowship. Fieldwork sample collection
by B. K. Place was supported by funding from Polar Knowledge Canada through the Northern
Scientific Training Program. Additional financial support for full redesign of the samplers was
provided through Environment and Climate Change Canada Grants & Contributions
(GCXE20S009). A. A. Colussi acknowledges support for this work through a Natural Sciences
and Engineering Research Council of Canada (NSERC) Graduate Scholarship – Master's program
(CGSM) and Ontario Graduate Scholarship (OGS). M. Lao acknowledges support for this work
through a NSERC Undergraduate Student Research Award (USRA). We thank C. M. Laprise and
C. Conlan for aid in the collection and organization of samples for analysis, supported in part by
the Memorial University Career Experience Program (MUCEP) and V. Sitahai through a York
University Dean's Undergraduate Research Award (DURA). T. C. VandenBoer, C. J. Young and
S. E. Ziegler were supported through the NSERC Discovery (RGPIN-2020-06166; RGPIN-2018-
05990; RGPIN-2018-05383) and Strategic Partnerships (479224) Programs. The authors would
also like to thank B. Hearn, D. Harris, A. Skinner, C. Young, J. J. MacInnis, J. Warren, and L.
Souza for their invaluable assistance in sampling site access and set up, off-season storage of
collection units, sample collection and analysis, and meteorological reanalysis. We thank H. Hung



and C. Shunthirasingham for productive discussions on modular design and considerations for collection of persistent pollutants.

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
