# Peer review of "Cost Effective Off-Grid Automatic Precipitation Samplers for"

_Atmospheric Measurement Techniques, 2023_

## Referee Comment (RC1)

This manuscript describes the development of an autonomous precipitation collector (excluding snow) that can be deployed off-grid in remote areas and is suitable for measuring pH, conductivity, and DOC (Dissolved Organic Carbon) in wet deposition with a monthly resolution. The collector was deployed and tested over a period of 2 years (2015-2016 on the NL-BELT network) for open fall and throughfall depositions.

**General comments:**

The subject is scientifically interesting and is well within the scope of the journal. But I think the presenting quality and arguments for validation of this system should be substantially improved before it can be accepted for publication in AMT:

The text is very dense, overly detailed, and often confusing, even off-topic, making it challenging to understand and preventing a clear presentation of the study's results. Moreover, certain parts of the text resemble a promotional brochure (e.g., in the "general design advantages" section) or a technical manual (e.g., 3.2.1) that does not appear suitable for a scientific article. I have the impression that this is a study report that has simply been slightly reformatted for submission to AMT (without even sorting through the relevant or irrelevant information for the purpose of this paper). A scientific article about a new method/system should demonstrate that the developed tool meets the expectations and requirements set to address scientific questions, rather than being a comparison with commercial tools or a series of technical information. For example, in 3.2.1, it doesn't matter that commercial samplers consume 100 times more; what is relevant here is the autonomy of your device.

The "Results" section mixes the points about the system validation and the scientific data analysis. The results/discussion section could be divided into two parts for clarity:

- A section titled "Validation": I believe there would be a significant benefit in simplifying the argument and presenting, point by point, the performance parameters. The article aims to demonstrate that the developed collectors are suitable for autonomously measuring conductive rain deposition in remote areas, so the key points of this demonstration could be highlighted:

  1. Autonomous off-grid operations: low power consumption + opening conditions for the collection of wet deposition (time taken to open or close...)
  2. Representativeness of the conductive rain collector for deposition measurement: Define what constitutes conductive rain compared to "common" rain.
  3. Validation of measured chemical parameters: Verify the preservation of chemical species even after a month, including pH, conductivity, and DOC.

- A section titled "Application": to demonstrate the usefulness of the collector in identifying canopy effects, through a comparison of the obtained data with the literature and between OF and TF for pH, conductivity, and DOC.

In addition to presenting quality, the developed arguments do not justify the scientific quality of the proposed measurements. The validation of the system's operation and measurement is somewhat fragmented and, as such, not very convincing. Specifically, three points are missing in the developed arguments :

1. This is a rain collection system where the opening/closing is controlled by a resistive sensor which is activated from the conductivity of the rain. However, the operational range of the sensor is not specified (it triggers at 1 mohm.cm, it is the only information). Thus, there is no definition of what is being truly measured.
2. The time it takes for the system to open is also not mentioned. We know that the first seconds of rain contain strong concentrations, so if the opening occurs 30 seconds after the start of the rain, a significant amount of information will be lost.
3. The system is dedicated to measuring monthly fluxes of chemical species, but no information is provided on the preservation of the tested parameters (pH, conductivity, and DOC) between each sample retrieval."

**Specific comments**

**Introduction :**

The introduction is very general and is organized into different paragraphs discussing the importance of monitoring atmospheric deposition of various chemical species (nitrates/sulfates (L83-91), POP (L119-129), ON (L157-165)) that are not subsequently addressed in the study, as only pH, conductivity, and DOC are studied here. If these species need to be detailed, it should be done in the conclusion to demonstrate that the developed system could also be used to study them. Please refocus the introduction on aspects related to your study.

L148-151: The justification for developing this new collector is based on comparisons with commercial devices, mentioning the cost and difficulty of making measurements in remote areas. It would be interesting to mention that precipitation collectors have already been developed to address specific questions about atmospheric fluxes. Here are a few examples:

- Laquer, F. C.: Sequential precipitation samplers: A literature review, Atmos. Environ. A.-Gen., 24, 2289–2297, https://doi.org/10.1016/0960-1686(90)90322-E, 1990.
- Germer, S., Neill, C., Krusche, A. V., Neto, S. C. G., and Elsenbeer, H.: Seasonal and within-event dynamics of rainfall and throughfall chemistry in an open tropical rainforest in Rondônia, Brazil, Biogeochemistry, 86, 155–174, https://doi.org/10.1007/s10533-007-9152-9, 2007.
- Laurent, B., Losno, R., Chevaillier, S., Vincent, J., Roullet, P., Bon Nguyen, E., Ouboulmane, N., Triquet, S., Fornier, M., Raimbault, P., and Bergametti, G.: An automatic collector to monitor insoluble atmospheric deposition: application for mineral dust deposition, Atmos. Meas. Tech., 8, 2801–2811, https://doi.org/10.5194/amt-8-2801-2015, 2015.
- Brahney, J., Wetherbee, G., Sexstone, G. A., Youngbull, C., Strong, P., and Heindel, R. C.: A new sampler for the collection and retrieval of dry dust deposition, Aeolian Research, 45, 100600, https://doi.org/10.1016/j.aeolia.2020.100600, 2020.
- Audoux, T., Laurent, B., Desboeufs, K., Noyalet, G., Maisonneuve, F., Lauret, O., and Chevaillier, S.: Intra-event evolution of elemental and ionic concentrations in wet deposition in an urban environment, Atmos. Chem. Phys., 23, 13485–13503, https://doi.org/10.5194/acp-23-13485-2023, 2023.

**2.1.2 Heated Precipitation Sensor**

Could you add information about time and delay between resistivity sensor activation and opening of the tip?

**3.0 Results and Discussion**

The paragraph begins with a summary of the strategies used for validation. Wouldn't it be better to place these points at the beginning of each section to avoid redundancy?

**3.1. General design advantage**

L518-543: This entire section mentions the advantages of the system without any results validating these claims. For example, regarding sample preservation, have you conducted tests on the preservation of reactive, volatile, or biologically transformed species (L538)? Similarly, you argue the possibility of detecting ultra-traces through the resistivity sensor systems (collection even at low conductivity and avoidance of dilution) and the types of materials used. I have no doubt that low-conductivity rains are collected, but what about the quality of the samples in the container? Could you provide information on field blanks, for example?

I believe that if you have a 'Validation' section, this part should be supported by your data.

**3.3. Comparison of Sample Collection Volumes**

L647: It is announced that the automatic collectors and total deposition collectors are colocated, but if I understand correctly, the values discussed in this paragraph only concern the volumes obtained with the total deposition collectors in OF?

L653 - 705: I don't understand your strategy. Why validate total deposition and then wet deposition, since the idea is to validate rain collectors? I question the need for an entire paragraph on the comparison of total deposition in the text with ECCC and DAYMET data. Comparing with model data at a km2 scale is interesting to show the representativity of measured fluxes at a single point on a larger scale (e.g., watershed scale), but it does not demonstrate that your collectors effectively capture the totality of precipitation. This study does not seem relevant to the paper's topic. I believe it could be included as supplementary material.

In my opinion, the comparison between total deposition and collector values (and their differences) is the best argument to show that the automatic collectors effectively capture all rainfall. The question is what is the representativity of the "conductive" rains collected here compared to the conventional rainfall collection?

L733-753: The RSD obtained on the triplicates of the rain collectors of 10 samples out of 33, or 1/3 of the samples, have an RSD greater than 40%. This indicates a very high variability in the collected volumes. It would be interesting to discuss the reasons for this variability, which is crucial to estimate the performance/artifacts of your device (it's unfortunate that the data are in supplementary material). No analysis is done on these high RSD values. Is it for the same period of the year, e.g., when the winds are strongest? Is it at a specific site? Did the replicates use the same sensor, or is each system independent of the other? Could the observed variability in the replicates be due to different closing or opening times? Has simultaneity been tested?

The comparison between total volumes and precipitation volumes shows a clear difference. The explanations given for these differences are not quantified, while they could provide information on the quality of the collection and, therefore, on the definition of conductive rains (and provide information on the dilution parameter). For example, is there a link between the total collected volume and the difference between total/conductive deposition? Is it related to

the site? This type of analysis could be done with scatterplots between total volume and rain, with different colors for each site to highlight if biases are site-related, then with colors by season.

Could you discuss these differences considering technical aspects? For example, the closing/opening of the collector lids is conditioned by the conductivity of the rainwater, assuming that concentrations in the rains decrease as the rain progresses. However, it has already been demonstrated that this is not necessarily the case (e.g. Audoux et al., 2023 see before). There can be refeeding of the lower layers with aerosols or mixtures of air masses that induce increases in conductivity during rain. The question is, what will be the behavior of the collector in this case? Is there a risk that the lid closes and reopens, or not? Are these phenomena that you observe from the datalogging?

It is known that the conductivity of a solution depends on temperature, and here the temperatures can vary greatly between seasons. The sensor is one of the key elements in the autonomy of your system. Have you tested how temperature affects the sensor's response? Could this have an influence on the differences observed between sites or with total depositions?

L718: The authors rely on acquired data that are not collected simultaneously with their samples and justify the discrepancies due to the heterogeneity of precipitation. However, this heterogeneity is known. Why choose to use ECCC measurement sites that are not collocated with NL-BELT sites to do this validation work for the collectors since it is certain that there will be a discrepancy between the two?

L733-776: This part pertains to the application of the collectors and no longer their validation; I think it should appear in a different section.

**3.4. Characterizing Chemical Parameters from NL-BELT**

I think the validation part of the chemical measurement should be in a separate section and thoroughly discuss:

Could you present sample preservation tests for pH, conductivity, and DOC? How can you ensure that concentrations are maintained over time despite all precautions taken? For example, have you taken a sample after rain and observed it after a month outdoors, considering temperature and sunlight variations (the choice of black color may lead to high temperatures inside the units during summer)? Could this have an impact on chemical parameters, such as the desorption of organic species from the surface of particles in the rain? The system is designed to limit evaporation, but is this really the case? Have you tested volumes pre- and post-sampling?

L995: This paragraph should be positioned earlier (in the validation section) to show the agreement between the values measured here with this new system and the values expected from the literature.

A significant portion of these results is applicable and pertains to the study of canopy effects on deposition fluxes and should be placed in a separate section and discussed accordingly

---

## Referee Comment (RC2)

**Cost Effective Off-Grid Automatic Precipitation Samplers for Pollutant and Biogeochemical Atmospheric Deposition**

Overview:

The paper outlines the design of an automatic precipitation sampler for off-grid use. It is suitable for measuring pH, conductivity, and dissolved organic carbon. The new sampler was tested in the Newfoundland and Labrador Boreal Ecosystem Latitudinal Transect over a two-year period for open-fall and throughfall precipitation. A notable disadvantage to this sampler is the inability to collect snowfall, limiting year-round monitoring of precipitation.

This work is of great interest to atmospheric scientists and is within the scope of the journal. Overall, this work is appropriate for acceptance in AMT following the revisions outlined below. The revisions mostly focus on reformatting, reducing lengthy descriptions, and improving clarity (particularly in the introduction and results/discussion sections).

Specific Comments:

**Introduction**

Overall, the introduction is very lengthy. Finding ways to pare down this material would allow for clarity of the important topics related to this work.

Line 74-91: It seems like this paragraph is trying to introduce pH, conductivity, and DOC, however it is hard to separate the information of these three topics from each other. The discussion of pH, conductivity, and DOC are mixed together, making it hard for the reader to parse out the relevant information for each topic. This paragraph needs to be revised and simplified.

Line 119-132: This paragraph discusses persistent organic pollutants (POPs), which the authors do not monitor during the testing period presented in this work. This paragraph could be removed from the introduction and incorporated later as a future use for the sampler.

Line 152-165: This paragraph discusses monitoring reactive nitrogen in atmospheric deposition, however this was not a focus of the precipitation characteristics (pH, conductivity, and DOC) that was highlighted in the results. If this does in fact connect with those three characteristics, those connections need to be made clearer, or this paragraph can be removed.

**Materials and Methods**

Line 305-306: Is the conductivity that triggers the sensor typically for that of precipitation (both in your geographic region and others)? I wonder about the variability of precipitation globally and if this would cause differences in sampling. In addition, what is the time delay for the lid opening once triggered?

**Results and Discussion**

Overall, I think the results and discussion could be reformatted. It was confusing to read and keep track of the sampler's validation results and how you were actually applying the sampler to gain new information (like TF and OF comparisons). I appreciate the "General Design Advantages" section to highlight the ability to use this sampler in remote locations and to collect replicates.

Line 681: If there is a lot of data missing from the ECCC monitoring site, why would you choose this dataset for your comparisons?

Line 689-690: The $R^2$ values are presented in a confusing manner, especially the ones found in parentheses. Please make it more clear which value corresponds to which site.

Overall, Section 3.3 Comparison of Sample Collection Volumes is very long and feels repetitive. This section should be made more concise.

Line 815-819: Mentioning and comparing the pH of the TF samples to the pH of the soil in that area seems like extraneous information. Why are you making this comparison? If it's truly needed, provide some justification or explanation as to why this is important.

Line 893-894: p-values to support your statements that the conductivity of $HgCl_2$ in water is comparable to field blanks and less than your samples would strengthen this argument.

Section 3.4.2 – Most other sections have a comparison of OF and TF samples, but this one does not. Was that intentional? A brief statement comparing these two would be great.

Line 958-963: You mention wildfire plumes being a potential cause of increased DOC levels in precipitation. Was there a wildfire event near your sites during your sampling period to warrant mentioning this possibility? Or would the possibility of increased DOC still be observed some amount of time after a wildfire event? In order to justify including this argument, I would like to see if you could make a potential correlation to an actual event that may explain this.

The discussion on DOC seems less than a validation of instrument performance, but rather a capability of the sampler. Unless you have DOC data from your sampling locations to compare to, this seems like new information for remote sampling sites.

Line 1014: what evidence do you have that supports the same that the automated system better maintains the integrity of DOC in samples?

Technical Corrections

Starting with section 3.4.3 – it is mentioned that flux (with the units: mg C $m^{-2}$ $d^{-1}$) is used. However, at several points in this discussion flux is given as mg C $m^{-2}$ $a^{-1}$. I'm providing a list locations I've found this mistake, but please check throughout the manuscript for places I may have missed:

Lines 50, 921, 925, 944, 955, 966, 997, 998, 1001

---

## Author Comment (AC1)

**General Response:**

We thank the Referees for their time to provide comments and feedback on this manuscript. We have edited the manuscript to address and incorporate their suggestions. In many cases, we realize that the clarity of our writing could be improved to address comments where the Reviewers had to make assumptions about technical details of our technique, and its validation. The Introduction has been thoroughly streamlined, as have sections of the Results and Discussion. We think that the manuscript contents are now clear, reflect the necessary corrections, and that the work is improved to a state we think meets the expectations for publication in Atmospheric Measurement Techniques. We hope the Editor and Reviewers agree. Our responses to specific comments from the Reviewers below are highlighted in ==yellow==, while additions and alterations to our manuscript or in cases where we are redirecting concerns to existing discussion are **highlighted, bolded, and underlined in green for additional clarity.**

**Editor**

This manuscript has been reviewed by two referees. They believe that the instrument you developed is of great interest and that the work is well conducted. Nevertheless, they both feel that the current manuscript contained a lot of redundant materials and should be substantially shortened. I agree with them very much; in addition, the structure of this manuscript should be further improved, as also pointed out by the two referees.

As a result, in addition to the revised manuscript and point-to-point reply, **could you please also provide a summary of major changes you make in the revision** (in order to make it easier for the referees and also the handling editor)? Thank you very much.

You - as the contact author - are requested to individually respond to all referee comments (RCs) by posting final author comments (ACs) on behalf of all co-authors no later than 28 March 2024 (final response phase).

We thank the Editor for their contribution to the discussion on changes required for this manuscript. We have collected the major points that required addressing from all parties to provide a succinct summary of the modifications to the revised manuscript. Listed below are the five major changes made to the manuscript:

1. The Introduction (Section 1.0) has been revised for clarity and substantially shortened to be better focused on the topic(s) of this manuscript. The following are the major modifications:
   a. In the original manuscript, the Introduction was 2,261 words total and the revised Introduction section is now 1,783 words total, a reduction by over $1/5^{th}$.
   b. Any discussion/mention of those chemical species (e.g., POPs, nitrates/sulfates, and ON) not specifically studied in this work has been moved and integrated succinctly into the new Conclusions and Future Directions (Section 4.0).
   c. Small – yet impactful – modifications were made to the preexisting information within the last paragraph of the Introduction to better addresses comments made by either Reviewer regarding the mixing of system validation and the scientific data analysis. The Authors believe that the location of these new modifications clarify and emphasize that sample preservation is based on established methodologies and that comparison to previously reported values in the literature is the primary metric for validation of the chemical measurements within this work.

2. A new subsection has been included within the Materials and Methods – more specifically, to the NL-BELT field site descriptions (Section 2.3).

   Sample Preservation (Section 2.3.1) has been added to the manuscript to address Reviewer comments regarding topics such as sample preservation and method validation.

   This is a re-working and expansion on the last paragraph in Section 2.3 of the original manuscript This subsection details our use of mercuric chloride ($HgCl_2$) as a sample preservation technique and cites its established efficacy in previous studies. With the well-established and long-studied nature of this preservation method, and its use for over a decade within the aquatic and soil biogeochemical work at NL-BELT, the Authors felt that additional tests were not needed to verify the temporal preservation efficacy of collected samples.

   We highlight in this new section the role of selective use of $HgCl_2$ within our replicate samplers as a method of internal sample validation (i.e. comparing samples in sterilized vs. unsterilized containers). We also highlight the versatility of our modular design as being open for use with many other established sample preservation methods, which ultimately allows it to become analyte specific without burdening researchers with laborious method re-development.

3. Within the Results and Discussion section comparing collected samples volumes (3.3), the paragraph detailing our intercomparison of ECCC and DAYMET measurements has been moved to the Supporting Information.

   This comparison can now be found in Section S2, "Deposition Comparison: This Work, DAYMET, and ECCC."

4. A rather minor, yet substantive, comment made regarding the paragraph in Section 3.3 discussing the presented triplicate sampler RSD values has been modified.

   The variability observed in the samples is now briefly revised and better addressed, with the preexisting sentences were reordered for better flow.

5. The paragraph examining cation exchange capacity has been completely removed from the Results and Discussion section analyzing precipitation pH within the collected NL-BELT samples (3.4.1). The Authors agree that it serves only our own scientific objectives at NL-BELT was unnecessary and tangential to the manuscript contents.

Reviewer 1

**Overview:**

This manuscript describes the development of an autonomous precipitation collector (excluding snow) that can be deployed off-grid in remote areas and is suitable for measuring pH, conductivity, and DOC (Dissolved Organic Carbon) in wet deposition with a monthly resolution. The collector was deployed and tested over a period of 2 years (2015-2016 on the NL-BELT network) for open fall and throughfall depositions.

**General comments:**

The subject is scientifically interesting and is well within the scope of the journal. But I think the presenting quality and arguments for validation of this system should be substantially improved before it can be accepted for publication in AMT:

The text is very dense, overly detailed, and often confusing, even off-topic, making it challenging to understand and preventing a clear presentation of the study's results. Moreover, certain parts of the text resemble a promotional brochure (e.g., in the "general design advantages" section) or a technical manual (e.g., 3.2.1) that does not appear suitable for a scientific article. I have the impression that this is a study report that has simply been slightly reformatted for submission to AMT (without even sorting through the relevant or irrelevant information for the purpose of this paper). A scientific article about a new method/system should demonstrate that the developed tool meets the expectations and requirements set to address scientific questions, rather than being a comparison with commercial tools or a series of technical information. For example, in 3.2.1, it doesn't matter that commercial samplers consume 100 times more; what is relevant here is the autonomy of your device.

We appreciate the thorough and thoughtful comments of the Reviewer. Where applicable, we have modified the text to be less dense, more concise, and less confusing for the readership. Given that there are no affordable commercial solutions for this type of sampling equipment, the manuscript needs to necessarily argue the rationale for the component selection and design. These sections aim to convey the unique features and advantages of our design, which is essential for assessment and effective uptake by the scientific community, particularly with respect to modular components. This is a central thrust of making instrumentation Open Access and research tools globally accessible.

We respectfully disagree with the position of the Reviewer on this point, as there are many manuscripts in this journal, and others related to instrument development, that are similarly detailed where novel platforms have been developed. As such, comparison with commercially available tools is relevant to underscore the application of our device in a broader context. When it comes to autonomy of using these samplers off-grid, power requirements are surely a fair comparison for stating why current commercial solutions are not viable.

The "Results" section mixes the points about the system validation and the scientific data analysis. The results/discussion section could be divided into two parts for clarity:

- A section titled "Validation": I believe there would be a significant benefit in simplifying the argument and presenting, point by point, the performance parameters. The article aims to demonstrate that the developed collectors are suitable for autonomously measuring conductive rain deposition in remote areas, so the key points of this demonstration could be highlighted:

  1. Autonomous off-grid operations: low power consumption + opening conditions for the collection of wet deposition (time taken to open or close...)
  2. Representativeness of the conductive rain collector for deposition measurement: Define what constitutes conductive rain compared to "common" rain.
  3. Validation of measured chemical parameters: Verify the preservation of chemical species even after a month, including pH, conductivity, and DOC.

We thank the Reviewer for their comments. We must note a major oversight on the part of the Reviewer and in our emphasis of method validation before proceeding further with our responses. Validation throughout this manuscript was conducted by controlling and observing in two ways:

i) Microbes will transform bioavailable molecules and so we microbially sterilized two out of three sample replicates with $HgCl_2$, which is standard practice for biogeochemical handling of environmental water samples for chemical characterization. This technique has been in use for over a decade within our aquatic and soil biogeochemical work at NL-BELT (Ziegler et al., 2017; Bowering et al., 2020; Bowering et al., 2022; Bowering et al., 2023). We then compare those sterilized results against a replicate that was not administered $HgCl_2$ to ascertain whether microbial issues exist for measurement metrics (e.g., conductivity, pH, DOC). Generally, we did not find a difference between measurements that were microbially sterilized versus those that were not for the presented measures, but this is not a given for other molecules. Given the longstanding history of this approach, we do not feel that extensive restructuring of the manuscript into the Reviewer's three sections is warranted.

We also strongly emphasize that this approach may not work for other molecules of interest to the atmospheric deposition community (e.g., those that are volatile). Sample preservation approaches should be conducted by users of this new platform based on their scientific objectives and review of the literature. We are not in a position to detail all preservation techniques. Please see our detailed response to comment 3 below and the newly added Section 2.3.1 entitled "Sample Preservation."

ii) We perform comparisons to the literature for validation of our measurements against longstanding measurement techniques used by academics and regulatory agencies globally. We do not see a scientific contribution that validates precipitation pH measurements as a worthwhile activity, given that the control experiments we have conducted to ensure our samples are preserved have been made in line with methodologies that have been in place since the 1960s.

We have addressed the numbered points offered by the Reviewer in detail as follows:

1. The time it takes for the system to open, based on several years of qualitative observation, has been added to the section entitled "Heated Precipitation Sensor" (2.1.2):

An output of 12 VDC is sent to the digital control board by the relay when rain is sensed, or 0 VDC in its absence, for signal processing and motor control (Figure S7). **When rain is sensed, the lid of each sampler in the array is simultaneously opened (<5 seconds) and is dependent on the rotational rate of the lid motor.** To increase the sensitivity of this sensor and to extend the sampling duration when conductive atmospheric constituents are completely washed out of the atmosphere, a sloped tin chute (e.g., Home Depot SKU# 1001110514) was added to extend the surface of the rain sensor.

2. For improved clarity, we will redefine what constitutes conductive rain compared to "common" rain in Section 3.1 (General Design Advantages):

The chute does this by accumulating water-soluble materials between rain events that require time to be completely washed off and through the release of ions from the material itself, which ages under environmental conditions. As the conductivity of the precipitation falls below the sensor threshold **– conductive precipitation being that which initially contains high solute levels that progress through trace level concentrations,** the added ions from the chute prolong the collection of rain past this time point. In rainfall events where extended atmospheric wash-out occurs, **where precipitation becomes ultrapure water,** the sampler lids will eventually close – preventing dilution of the sample while maintaining the collection of analytes of interest.

3. Section 2.3 (Continuous Monthly Collection of Remote Samples at NL-BELT) was revised to include a new 'Sample Preservation' subsection (Section 2.3.1). This subsection addresses our use of a well-established sample preservation technique and clarifies why additional tests were not done to verify the preservation of collected chemical species over time. We believe that this new subsection, in addition to the small modifications made to clarify the motivations and objective of our approach in the final paragraph of the Introduction addresses the comments made by both Reviewers regarding their initial confusion on the mixing of system validation and the scientific data analysis. Please see below for the modifications made to the Introduction as well as the new additions to Section 2.3.1 specifically pertaining to sample preservation and validation:

**1.0. Introduction (final paragraph)**

The materials used can be easily changed in order to optimize collection **and preservation** of a wide array of target analytes, such as DOC, **when using high density polyethylene and mercuric chloride (HgCl$_2$).** We demonstrate that these platforms are capable of continuous operation off-grid for monthly wet deposition collection of precipitation across the Newfoundland and Labrador Boreal Ecosystem Latitudinal Transect (NL-BELT) during snow-free periods in 2015 and 2016.

The captured fraction compared to total volume deposited is used to gain insight into how these samplers can limit analyte dilution effects and improve method detection limits**, such as rejecting 50% of the total volume delivered as ultrapure precipitation leading to a factor of two improvement**. Chemical parameters of pH, conductivity, and DOC fluxes **collected according to established preservation protocols** were then **compared to prior measurements** to validate this proof-of-concept system. Measurement methods for pH and conductivity of rainwater are very well-established in the literature and serve as a baseline reference to ensure that the samples collected by the new devices presented in this work are consistent with what is expected in samples from a remote coastal environment, given the selective sampling strategy. We then move away from these well-established parameters to quantify DOC fluxes **using established biogeochemical preservation techniques for fresh water and groundwate**r to demonstrate the potential of these samplers in application to automated collection of analytes of emerging importance and interest in the remote locations of our latitudinal transect.

**2.3.1 Sample Preservation**

Four of the six sample containers (two each of OF and TF) were biologically sterilized using 1 mL of a saturated aqueous solution of mercuric chloride (HgCl$_2$) to preserve against biological growth and loss of bioavailable nutrients over the collection periods. Unsterilized sample containers (without HgCl$_2$) were used for measurements of recalcitrant species and to assess any matrix effects exerted on target analyte quantitation. **The use of HgCl$_2$ as a sample preservation technique has been long-studied and well-established (Kirkwood, 1992; Kattner, 1999); thus, additional tests to verify the preservation of collected chemical species over time were not performed. The analysis of deposition collected in unsterilized and**

**sterilized containers, however, serves as a method for internal sample validation - as does our evaluation of measurement outcomes in comparison to those reported within the literature.**

Filtered samples were transferred to new clean HDPE containers and stored for up to two months at 4°C in a cold room until analysis. **The target analytes in this work are non-volatile and the described sample collection methods consider this analyte property, as well as their interactions with container materials. The versatility of the system design allows for the use of different collection materials, keeper solvents for volatile organics, etc., so that the experimental design can be analyte specific, depending on end user needs. Sample preservation approaches should thus be identified by users of this new platform based on their scientific objectives and review of the literature (Galloway and Likens, 1978; Peden et al., 1986; Dossett and Bowersox, 1999; Wetherbee et al., 2010). In addition to the internal validation approach described here, we aim to demonstrate that the precipitation samplers in this work are suitable for measuring conductive deposition on- and off-grid. Below we highlight autonomous off-grid operations, determine the fraction of conductive rainfall collected from the total volume of precipitation, and validate our measurements through comparison to the literature.**

A section titled "Application": to demonstrate the usefulness of the collector in identifying canopy effects, through a comparison of the obtained data with the literature and between OF and TF for pH, conductivity, and DOC.

We thank the Reviewer for their suggestion. We believe that our existing Sections 3.3 and 3.4 demonstrate the 'Applications' of our collector in identifying canopy effects as well as comparing our measurements to those reported within the relevant literature. The extent of this analysis serves as an example of how the samplers may be used, which will be exploited in our future work to explicitly consider canopy effects on deposited species. For brevity, as both Reviewers and the Editor have requested reduced content, we have opted not to separate and expand the discussion of canopy effects with respect to the observed OF and TF deposition.

Instead, we have tried to emphasize this in our Conclusions and Future Directions (Section 4.0):

Comparability and complementariness of our results to well-established and current measurements of interest like DOC, demonstrate their efficacy **and potential application to the study of processes such as canopy-precipitation interactions through the collection of open fall and throughfall replicates.**

**The samples collected in this work from this new instrumentation, specifically, are expected to be used further in several upcoming complementary and novel environmental monitoring studies. Not only will this future work extend our biogeochemical analysis, but it will also assist in our studying of the transport of other anthropogenic pollutants of emerging interest which are beyond the scope of describing this new platform.**

In addition to presenting quality, the developed arguments do not justify the scientific quality of the proposed measurements. The validation of the system's operation and measurement is somewhat fragmented and, as such, not very convincing. Specifically, three points are missing in the developed arguments:

1. This is a rain collection system where the opening/closing is controlled by a resistive sensor which is activated from the conductivity of the rain. However, the operational range of the sensor is not specified (it triggers at 1 mohm.cm, it is the only information). Thus, there is no definition of what is being truly measured.

We thank the Reviewer for highlighting this point, but it is also somewhat confusing. A threshold for generating a response would exist in all sensors, so the one we have provided represents the lower limit of the range. This is common practice for many instrumental techniques, where the detection limit is reported. We will note that all standard deposition samplers in government monitoring networks use the same sensing approach for precipitation (often without a design to increase their sensitivity; e.g., NADP, CAPMoN, etc.; (Canadian Air and Precipitation Monitoring Network, 1985b)). Without a specific reference to other programs' definitions, and some sort of example expectation from the Reviewer, is it hard to ascertain what might be most useful to address their concern. As such, we have retained our original lower limit conductivity definition and added the equivalent concentration in sodium chloride. We think this clarifies the work and hope the Reviewer agrees.

We have added the following to Section 2.1.2, "Heated Precipitation Sensor":

The detection of rain modulates the opening and closing of the collection units by an interdigitated resistive sensor (M152; Kemo Electronic GmbH, Geestland, Germany; Figures S6 to S8). **This approach is consistent with established precipitation detection techniques used by government monitoring programs (e.g., CAPMoN; Canadian Air and Precipitation Monitoring Network, 1985a, 1985b).** The rain sensor detects conductive deposition by the completion of a conductive circuit when electrolytes bridge the connection between the interdigitated gold electrodes. The sensor is supplied with 12 VDC from the power system to trigger a relay when precipitation conductance above 1 MΩ·cm conductivity is detected (determined experimentally, see Section S1). **This is equivalent to approximately 8 μM sodium chloride. The sensor detection limit reflects an upper limit of precipitation ion loading because the design of the rain chute leads to an increase in surface area of more than a factor of 25 on which solutes can accumulate to enhance the ionic content of the deposited water.** An output of 12 VDC is sent to the digital control board by the relay when rain is sensed, or 0 VDC in its absence, for signal processing and motor control (Figure S7).

2. The time it takes for the system to open is also not mentioned. We know that the first seconds of rain contain strong concentrations, so if the opening occurs 30 seconds after the start of the rain, a significant amount of information will be lost.

The Reviewer mentioned this point in a prior comment, which has already been addressed with a technical addition to the paper. We thank the Reviewer for this comment, and we understand the need to communicate that the samplers open readily when rain is sensed, for the facts they mention here. The opening of the lid is fast (<5 seconds) and is dependent on the rotation rate of the motor selected. We typically use 2 to 6 rpm motors, depending on their availability from our suppliers. We hope the inclusion of this information in our revision is satisfactory.

3. The system is dedicated to measuring monthly fluxes of chemical species, but no information is provided on the preservation of the tested parameters (pH, conductivity, and DOC) between each sample retrieval."

We agree with the Reviewer, and we have now included a new subsection (Section 2.3.1) to address our use of a well-established sample preservation technique for microbially labile DOC, and hence, why additional tests were not done to verify the preservation of collected chemical species over time. We have also now clearly acknowledged that other analytes may require other considerations, depending on their chemical and physical properties (e.g., volatility).

We have also noted in our responses above that our comparison to long-standing reports of a parameter such as precipitation pH, using calibrated meters and standard collection/storage/analysis techniques from the literature (Dossett and Bowersox, 1999; Wetherbee et al., 2010) means that further evaluation of the preservation is not warranted, as the outcomes (i.e., the samples will be stable) are predictable. Given that both Reviewers have also indicated that the manuscript is too long, we do not feel that adding more results to this work are warranted. They have taken the time and attention to identify redundant components of the original work and we feel that it is important to emphasize where we find suggested additions to also be redundant. Especially results that fail to make substantive contributions to this field. We hope that our efforts to improve clarity in our justified approach, guidance for the community, and doing so concisely for the overall improvement of the manuscript are agreeable and meaningful changes to address the concerns of the Reviewer.

**Specific comments:**

**Introduction**

The introduction is very general and is organized into different paragraphs discussing the importance of monitoring atmospheric deposition of various chemical species (nitrates/sulfates (L83-91), POP (L119-129), ON (L157-165)) that are not subsequently addressed in the study, as only pH, conductivity, and DOC are studied here. If these species need to be detailed, it should be done in the conclusion to demonstrate that the developed system could also be used to study them. Please refocus the introduction on aspects related to your study.

We thank the Reviewer for this comment and Reviewer 2 raised similar concerns. The Introduction has been revised and the mention/discussion of studying the deposition of various chemical species (POPs, ON, etc.) not specifically studied in this work has been moved and integrated into the newly titled Section 4.0, "Conclusions and Future Directions" (see below). In the original manuscript, the Introduction was 2,261 words total, and the revised Introduction section is now 1,783 words total.

**4.0 Conclusions and Future Directions (last paragraph):**

For the broader deposition-motivated community, the instrument design also allows for easy cost-effective modification of the number of replicate samplers, the material composition of all surfaces the aqueous samples interact with, as well as preservation strategies - depending on the analyte of interest. **For example, the lack of organic nitrogen measurements within universally established sampling and measurement procedures serves as a general example of the substantial knowledge gaps that may result when translating limited data sets to the wider global picture. This includes incomplete speciation and quantification across precipitation, aerosol, and gas phases. Monitoring systems that support U.S. deposition assessments (e.g., the NADP) only characterize the inorganic fraction of wet deposition. Additionally, modern emerging issues that require the continuation of existing deposition measurements or expansion of observation programs revolve around identifying and quantifying compound classes of concern, such as persistent organic pollutants. As reported in the literature, the deposition of these types of pollutants (e.g., polychlorinated biphenyls, polycyclic aromatic hydrocarbons, etc.) can be monitored using suitable collectors made of amber-coloured glass or stainless steel (Fingler et al., 1994; Amodio et al., 2014) - modifications which can be applied to the sample design detailed here. The samples collected in this work from this new instrumentation, specifically, are expected to be used further in several upcoming complementary and novel environmental monitoring studies. Not only will**

**this future work extend our biogeochemical analysis, but it will also assist in our studying of the transport of other anthropogenic pollutants of emerging interest which are beyond the scope of describing this new platform.**

L148-151: The justification for developing this new collector is based on comparisons with commercial devices, mentioning the cost and difficulty of making measurements in remote areas. It would be interesting to mention that precipitation collectors have already been developed to address specific questions about atmospheric fluxes. Here are a few examples:

- **Laquer, F. C**.: Sequential precipitation samplers: A literature review, Atmos. Environ. A.-Gen., 24, 2289–2297, https://doi.org/10.1016/0960-1686(90)90322-E, 1990.
- **Germer, S**., Neill, C., Krusche, A. V., Neto, S. C. G., and Elsenbeer, H.: Seasonal and within-event dynamics of rainfall and throughfall chemistry in an open tropical rainforest in Rondônia, Brazil, Biogeochemistry, 86, 155–174, https://doi.org/10.1007/s10533-007-9152-9, 2007.
- **Laurent, B**., Losno, R., Chevaillier, S., Vincent, J., Roullet, P., Bon Nguyen, E., Ouboulmane, N., Triquet, S., Fornier, M., Raimbault, P., and Bergametti, G.: An automatic collector to monitor insoluble atmospheric deposition: application for mineral dust deposition, Atmos. Meas. Tech., 8, 2801–2811, https://doi.org/10.5194/amt-8- 2801-2015, 2015.
- **Brahney, J**., Wetherbee, G., Sexstone, G. A., Youngbull, C., Strong, P., and Heindel, R. C.: A new sampler for the collection and retrieval of dry dust deposition, Aeolian Research, 45, 100600, https://doi.org/10.1016/j.aeolia.2020.100600, 2020.
- **Audoux, T.**, Laurent, B., Desboeufs, K., Noyalet, G., Maisonneuve, F., Lauret, O., and Chevaillier, S.: Intra-event evolution of elemental and ionic concentrations in wet deposition in an urban environment, Atmos. Chem. Phys., 23, 13485–13503, https://doi.org/10.5194/acp-23-13485-2023, 2023.

We thank the Reviewer for directing our attention to these overlooked studies. We have added these in locations where we think they best fit in the manuscript.

1. We mention two of the five references listed above (Laquer, 1990; Germer, 2007), amongst others, within a modified Introduction paragraph to better address the preexisting precipitation collectors that have been more broadly developed for a variety of scientific objectives:

**Over the past 60 years, the precipitation chemistry community has made advancements in deposition collectors to better understand atmospheric processes (Siksna, 1959).** While bulk deposition collection (i.e., a collection bucket or jug fitted with a funnel open at all times; Hall, 1985) is both a simple and economically feasible sampling method utilized by monitoring networks, it is subject to bias through collection of inputs other than atmospheric deposition (e.g., bird droppings, insects, plant debris). As a result, bulk collectors can overestimate total deposition and underestimate wet deposition in a variety of locations (Lindberg et al., 1986; Richter and Lindberg, 1988; Stedman et al., 1990). **Sequential precipitation collection methods include manually segmenting samplers (requiring only a shelter, clean surface, and an operator), linked collection vessels (sample containers that are filled in sequence via gravitational flow), amongst others and have been developed to analyze rainwater composition and measure parameters such as pH and conductivity (Gatz et al., 1971; Reddy et al., 1985; Vermette and Drake, 1987; Laquer, 1990). Sequential sampler designs have also been adapted to collect precipitation in remote field sites (Germer et al., 2007; Sanei et al., 2010).** Although it is a more costly and time intensive method when compared to bulk deposition collection, the major appeal of measuring isolated wet deposition is **the ability to isolate this individual atmospheric process.**

2. We have briefly mentioned three references (Laurent et al., 2015; Brahney et al., 2020; Audoux et al., 2023), in addition to Germer et al., 2007 in Section 3.1 (General Design Advantages) rather than in the Introduction. The point of precipitation samplers being designed to address specific scientific objectives is a segue into the existing need for innovation geared towards more versatile sample collection:

**While several precipitation collectors have been similarly developed to address specific scientific objectives – e.g., the quantitation of dust in wet and dry deposition (Laurent et al., 2015; Brahney et al., 2020), determination of ions and DOC in a tropical rainforest (Germer et al., 2007) and urban environments (Audoux et al., 2023), here we present a more general design for modular adaptability.** When compared to other precipitation collection apparatuses, the automated precipitation sampler developed in this work has several advantages. Most notable is the ability to collect integrated samples at remote locations by exploiting its off-grid capabilities. ... In rainfall events where extended atmospheric wash-out occurs, where precipitation becomes ultrapure water, the sampler lids will eventually close – preventing dilution of the sample while maintaining the collection of analytes of interest. **A recent study has found that rainfall events could exhibit variability and the lower atmosphere can be supplied with aerosols due to specific sources, atmospheric dynamics, and meteorological conditions (Audoux et al., 2023). If this occurs, the automated lid will reopen to sample the polluted air masses.** In application to trace pollutants, this also reduces methodological sample preparation time as it decreases the extent to which additional handling steps, like solid-phase extraction, are required prior to analytical determinations.

**2.1.2 Heated Precipitation Sensor**

Could you add information about time and delay between resistivity sensor activation and opening of the tip?

We thank the Reviewer for this suggestion, and we understand the need to include this information within the manuscript. As the Reviewer has mentioned this point in prior comments, it has been already addressed with a technical addition to the paper (see Section 2.1.2, "Heated Precipitation Sensor").

**Results and Discussion**

**3.0 Results and Discussion**

The paragraph begins with a summary of the strategies used for validation. Wouldn't it be better to place these points at the beginning of each section to avoid redundancy?

We thank the Reviewer for their comment. Due to the detailed nature of Section 3.0, we thought including this summary would effectively preface each of the following subsections. Additionally, with these important details summarized at the beginning of the Results and Discussion section, this benefits those readers who would choose only to skim through the paper to points of particular interest to them. As such, we have opted to retain this section.

**3.1 General Design Advantage**

L518-543: This entire section mentions the advantages of the system without any results validating these claims. For example, regarding sample preservation, have you conducted tests on the preservation of reactive, volatile, or biologically transformed species (L538)? Similarly, you argue the possibility of detecting ultra-traces through the resistivity sensor systems (collection even at low conductivity and avoidance of dilution) and the types of materials used. I have no doubt that low-conductivity rains are collected, but what about the quality of the samples in the container? Could you provide information on field blanks, for example?

I believe that if you have a 'Validation' section, this part should be supported by your data.

We thank the Reviewer for their comment. This section was addressed in response to previously made Reviewer comments via a technical addition to the paper (see Section 2.3.1., Sample Preservation).

With respect to field blanks, we have an existing statement on how these were prepared in Section 2.4 'Cleaning and Preparation of Sample Containers' and have added a minor clarification that these were used to perform blank subtractions on measurements of all relevant metrics (e.g., conductivity and DOC here):

Field and method blanks were collected through the addition of DIW to cleaned containers, and/or sample handling devices, in order to quantify appropriate method detection limits and to identify any sources of systematic or random contamination. **Blank subtraction was applied to measurements, where appropriate.**

An example of an appropriate mention of how field blanks were used for considering conductivity (Section 3.4.2, first paragraph):

The conductivity of saturated $HgCl_2$ in water (at 0.1% vol/vol) was $13.6 \pm 0.4$ µS/cm, which is also comparable to **but statistically higher than** our field blanks **(p = 0.0015; unpaired t test)** and less than what was observed for our samples **($p < 2 \times 10^{-6}$ for each site considered separately and also across all sites; unpaired t-test).**

**3.3. Comparison of Sample Collection Volumes**

L647: It is announced that the automatic collectors and total deposition collectors are colocated, but if I understand correctly, the values discussed in this paragraph only concern the volumes obtained with the total deposition collectors in OF?

The Reviewer understands correctly that this paragraph deals with the fraction of total deposition collected in the automated samplers. This paragraph demonstrates the extent to which atmospheric washout can potentially dilute bulk samples, making the determination of trace analytes challenging and often then subject to complex and error-prone sample concentration methods. We refer the Reviewer to the existing introdcutory text in the Methods (Section 2.3) describing the colocation of the samplers:

One array of six automated collection units (3 OF, 3 TF) were deployed within one forested experimental field site located in each of the four watershed regions of the NL-BELT (24 samplers in total) between 2015 and 2016. Additionally, between one to three total deposition samplers were located at each of the four field sites (Table 1, Figure S10).

In Section 3.3 itself we have clarified the comparators in the second paragraph, alongside expectations:

The wet deposition volumes collected for the snow free period using the automated precipitation samplers did not follow the trends in total deposition (Figure 4), as might be expected **(e.g., due to pollutant loading, rainfall quantity/rate, and scavenging processes).** For the 2015 collection period from June through October, the summed volumes of OF precipitation, from south to north across the NL-BELT, were 25.4, 10.9, 20.4, and 2.2 L, while in 2016 they were 17.3, 30.4, 13.5, and 5.1 L. **While the total and OF fractions would typically be much closer to unity in more polluted regions, it would be expected in these remote NL-BELT field sites for the differences to be driven by complex, non-linear processes that cannot be easily disentangled. Here we present** three reasons as to why the measured wet **OF** deposition volumes do not follow the total deposition trend across the transect.

L653 - 705: I don't understand your strategy. Why validate total deposition and then wet deposition, since the idea is to validate rain collectors? I question the need for an entire paragraph on the comparison of total deposition in the text with ECCC and DAYMET data. Comparing with model data at a km2 scale is interesting to show the representativity of measured fluxes at a single point on a larger scale (e.g., watershed scale), but it does not demonstrate that your collectors effectively capture the totality of precipitation. This study does not seem relevant to the paper's topic. I believe it could be included as supplementary material.

We thank the Reviewer for their comment. We agree with the suggestion that this comparison (L673-701 in the original manuscript) be removed and we have relocated it to the Supporting Information. This significantly simplifies and focuses this section and we thank the Reviewer for encouraging us to do so with this comment.

The text in Section 3.3 of the manuscript now simply refers to this material, presenting only the most important results of the comparison:

The automated samplers were collocated with total deposition samplers and deployed across the experimental forests of four NL-BELT regions during the 2015 and 2016 growing seasons to observe deposition trends. In addition, we briefly compare these observations to the long-term climate normals reported by ECCC and estimated deposition at 1 km x 1 km resolution from the DAYMET reanalysis model (Table 1, **Section S2)**. Three automated samplers were deployed in the open to collect incident precipitation (OF) and another three under the experimental forest site canopy (TF). ... The total deposition samplers were installed in HR in late 2014 and the automated samplers were then set up at PB. Despite this, there is good agreement between the trends in predicted deposition values by DAYMET with the measured values, although the absolute amounts from these are systematically lower in all of our observations **(Section S2)** Regardless, by following the recommended siting criteria from the NADP and CAPMoN as best as possible, the very strong agreement of our temporal trends at both annual and monthly timescales with both comparators demonstrates the suitability of the total deposition samplers and, therefore, the automated samples for use in quantifying deposited chemical species of atmospheric interest into the experimental sites.

In my opinion, the comparison between total deposition and collector values (and their differences) is the best argument to show that the automatic collectors effectively capture all rainfall. The question is what is the representativity of the "conductive" rains collected here compared to the conventional rainfall collection?

We thank the Reviewer for this comment. We assume that the conventional rainfall referred to by the Reviewer is bucket-based bulk deposition, collected on an event basis, rather than integrated sampling. If this is the case, our conductive rain is the fraction containing solutes above the sensor detection threshold stated in the main manuscript (addressed previously, see above).

The representativeness is exactly that defined by the sensor threshold, excluding ultrapure water from precipitation during washout events, which serves to dilute analytes in a sample. In more polluted regions, for example, we would expect the total and open fall fractions to be much closer to unity (see response and manuscript additions made to the comment immediately before this one). At the remote locations of the NL-BELT experimental forests - knowing that the region is subject to intense rainfall - it is not surprising that the collected fractions are smaller than found in the total deposition samplers. The fact that we do not see any comparisons where open fall volume exceeds that of the total also satisfies the basic premise of our comparison.

L733-753: The RSD obtained on the triplicates of the rain collectors of 10 samples out of 33, or 1/3 of the samples, have an RSD greater than 40%. This indicates a very high variability in the collected volumes. It would be interesting to discuss the reasons for this variability, which is crucial to estimate the performance/artifacts of your device (it's unfortunate that the data are in supplementary material). No analysis is done on these high RSD values. Is it for the same period of the year, e.g., when the winds are strongest? Is it at a specific site? Did the replicates use the same sensor, or is each system independent of the other? Could the observed variability in the replicates be due to different closing or opening times? Has simultaneity been tested?

We thank the Reviewer for their comment, and we agree that the high RSD values should be addressed better. To do so, we will answer your questions posed individually as follows:

    i.    Is it for the same period of the year, e.g., when the winds are strongest?

       Winds are highly variable throughout the year. Storms can be accompanied by winds up to 140 km/hr in any month. Siting of samplers was conducted according to standard requirements of government sampling statements, so subject to the same potential artifacts. Government agencies tend to only collect a single sample, so potential uncertainty in volume and analyte deposited quantities likely goes unreported (see addition to Section 3.3. below).

ii.    Is it at a specific site?

No; it occurs at many of the sites and varies over the sampling periods.

iii.    Did the replicates use the same sensor, or is each system independent of the other? Could the observed variability in the replicates be due to different closing or opening times? Has simultaneity been tested?

As described in Sections 2.1.2 and 2.3, each array of 6 samplers (3 OF, 3 TF) are controlled by the same sensor and the sampler lids all open within 5 seconds of detected conductive rainfall. Thus, after several years of qualitative observations, the observed variability is not a result of different sampler lid opening and closing times (i.e., a lack of simultaneity).

The following modifications have been made to Section 3.3 in the manuscript to communicate the information we have provided above for questions (i) and (ii). In addition to the new text underlined and bolded, some sentences from the original manuscript were reordered in this modified paragraph for better flow:

The automated OF wet deposition volumes collected each year have peak values that range from 1 to 4 L with an overall variability of ± 33% for any triplicate of samples across the entire dataset. Across our 33 sample collection periods, our replicate relative standard deviations (RSDs) follow a log-normal distribution where volume reproducibility is typically within ± 12.5% and almost always within ± 31.5% (Figure S11). A few outliers with higher variability skew the overall view of volume precision. Out of 33 OF samples collected, **10** have RSDs greater than 40.5% with 2 of those **10** having RSDs greater than 100%. **Those values greater than 40.5% had no systematic relationship with site or time of year**. **Wind speeds were considered as a possible source of variability. The prevailing winds over Atlantic Canada are known to be southwesterly in the summer – intensifying during the autumn months – and westerly to northwesterly in the winter (Bowyer, 1995; Jacob, 1999; Randall, 2015). Strong wind speeds (i.e., >100 km/hr) could occur on an event basis during any time of the year and, thus, could contribute to the variability seen at each field site.** Wind is known to generate bias in gauge-based precipitation measurements where unshielded precipitation gauges can catch less than half of the amount of a shielded gauge (Colli et al., 2016). A windscreen design for obtaining rainfall rates – and thus, volumes – to be more reproducible could be considered in future deployments of our developed samplers, similar to recently reported innovations for smaller rainfall rate devices (Kochendorfer et al., 2023). This would, however, increase costs and logistical considerations in deploying the developed devices which currently operate synonymously to deposition systems employed by government monitoring programs. **Our siting approach is consistent with these, which often deploy a single sampler without wind protection. Thus, by employing replicates, we are able to better ascertain the environmental variability.** In addition, collection of replicate samples allows our observations to span a wider physical area, reducing the impact of confounding variables such as wind speed in comparison to a more typical sample size of one for many field collections. Imperfect siting and lack of shielding is necessary where remote field sampling prevents the setup of such infrastructure. As a result, the deployment of triplicate samplers provides researchers with a better opportunity to implement quality control as they can reduce bias in the event of dynamic OF. While the effect of wind is reduced, additional factors can drive variability when the samplers are placed under a forest canopy for TF collection.

The comparison between total volumes and precipitation volumes shows a clear difference. The explanations given for these differences are not quantified, while they could provide information on the quality of the collection and, therefore, on the definition of conductive rains (and provide information on the dilution parameter). For example, is there a link between the total collected volume and the difference between total/conductive deposition? Is it related to the site? This type of analysis could be done with scatterplots between total volume and rain, with different colors for each site to highlight if biases are site-related, then with colors by season.

Similar to our response to a comment above, the difference between open fall and total volume is not simple (e.g., pollutant loading, rainfall quantity/rate, scavenging processes). It is generally true that in more polluted regions, we would expect the total and open fall fractions to be much closer to unity. However, at the remote locations of the NL-BELT experimental forests, we expect the differences (as shown in Figure 4) to be driven by complex, non-linear processes that cannot be easily disentangled.

To better communicate this point, we have made the following modifications in Section 3.3:

The wet deposition volumes collected for the snow free period using the automated precipitation samplers did not follow the trends in total deposition (Figure 4), as might be expected **(e.g., due to pollutant loading, rainfall quantity/rate, and scavenging processes).** For the 2015 collection period from June through October, the summed volumes of OF precipitation, from south to north across the NL-BELT, were 25.4, 10.9, 20.4, and 2.2 L, while in 2016 they were 17.3, 30.4, 13.5, and 5.1 L. **While the total and OF fractions would typically be much closer to unity in more**

**polluted regions, it would be expected in these remote NL-BELT field sites for the differences to be driven by complex, non-linear processes that cannot be easily disentangled.** Here we present three reasons as to why the measured wet deposition volumes do not follow the total deposition trend across the transect.

Could you discuss these differences considering technical aspects? For example, the closing/opening of the collector lids is conditioned by the conductivity of the rainwater, assuming that concentrations in the rains decrease as the rain progresses. However, it has already been demonstrated that this is not necessarily the case (e.g. Audoux et al., 2023 see before). There can be refeeding of the lower layers with aerosols or mixtures of air masses that induce increases in conductivity during rain. The question is, what will be the behavior of the collector in this case? Is there a risk that the lid closes and reopens, or not? Are these phenomena that you observe from the datalogging?

We thank the Reviewer for this question. We acknowledge that the lower layers of the atmosphere can be supplied with aerosols, or that the arrival of contaminated air masses can increase the conductivity during a washout event. In that case, the lid of our samplers would re-open and sample the conductive precipitation, as the sensor is always active and the discriminating factor between sample collection (or not). We have witnessed this phenomenon firsthand in the field during deployment. To better address this topic, we have modified Section 3.1 to include the following underlined text:

In rainfall events where extended atmospheric wash-out occurs, where precipitation becomes ultrapure water, the sampler lids will eventually close – preventing dilution of the sample while maintaining the collection of analytes of interest. **A recent study has found that rainfall events could exhibit variability and the lower atmosphere can be supplied with aerosols due to specific sources, atmospheric dynamics, and meteorological conditions (Audoux et al., 2023). If this occurs, the automated lid will reopen to sample the polluted air masses.** In application to trace pollutants, this also reduces methodological sample preparation time as it decreases the extent to which additional handling steps, like solid-phase extraction, are required prior to analytical determinations.

It is known that the conductivity of a solution depends on temperature, and here the temperatures can vary greatly between seasons. The sensor is one of the key elements in the autonomy of your system. Have you tested how temperature affects the sensor's response? Could this have an influence on the differences observed between sites or with total depositions?

We thank the Reviewer for highlighting this point. The sensor surface is heated during detected precipitation events, which would mitigate the majority of ambient temperature effects. The operational temperature range provided by the manufacturer, based on a personal communication with their support team, is -10 °C to +55 °C. Based on many years of qualitative observations, we have not noticed seasonal temperatures influencing the response of the sensor and know that similar considerations are not made by commercial systems on the market, or by government agency samplers. Lastly, the automated deposition samplers were decommissioned during the winter at the NL-BELT so we felt that this information was not relevant to include in the main text. It is noteworthy to mention that we have deployed these automated samplers year-round, since this initial study, in temperatures below –10 °C in Toronto and they continue to detect snow and ice deposition without issue. To do so, we have heated our chute to melt any deposited snow or ice, which again prevents any temperature-dependent conductivity detection issues. This will be featured in future studies reporting on winter deposition of pollutants of interest.

In addition, we have included the operating temperature range provided by the manufacturer of **(-10°C to +55°C)** in Table S2 within the Supporting Information.

L718: The authors rely on acquired data that are not collected simultaneously with their samples and justify the discrepancies due to the heterogeneity of precipitation. However, this heterogeneity is known. Why choose to use ECCC measurement sites that are not collocated with NL-BELT sites to do this validation work for the collectors since it is certain that there will be a discrepancy between the two?

We thank the Reviewer for highlighting this. We agree with their assessment and, as described above, have moved this comparison to the Supporting Information.

L733-776: This part pertains to the application of the collectors and no longer their validation; I think it should appear in a different section.

We thank the Reviewer for their comment. As there is now a separate "Sample Preservation" subsection (2.3.1), we believe that this particular analysis can remain where it is within the main text.

**3.4. Characterizing Chemical Parameters from NL-BELT**

I think the validation part of the chemical measurement should be in a separate section and thoroughly discuss:

Could you present sample preservation tests for pH, conductivity, and DOC? How can you ensure that concentrations are maintained over time despite all precautions taken? For example, have you taken a sample after rain and observed it after a month outdoors, considering temperature and sunlight variations (the choice of black color may lead to high temperatures inside the units during summer)? Could this have an impact on chemical parameters, such as the desorption of organic species from the surface of particles in the rain? The system is designed to limit evaporation, but is this really the case? Have you tested volumes pre- and post-sampling?

The Reviewer mentioned this point in a prior comment, which has already been addressed with a technical addition to the paper (Section 2.3.1, Sample Preservation). This subsection addresses our use of a well-established sample preservation technique and hence, why additional tests were not done to verify the preservation of collected chemical species over time. We believe that this new subsection addresses all further comments made by either Reviewer regarding system validation, as do the references we have provided, should they have further interest in this topic.

L995: This paragraph should be positioned earlier (in the validation section) to show the agreement between the values measured here with this new system and the values expected from the literature.

We respectfully disagree with the Reviewer in this case. After the thorough restructuring of the manuscript, we believe this paragraph is most effective in its current position.

A significant portion of these (Section 3.4) results is applicable and pertains to the study of canopy effects on deposition fluxes and should be placed in a separate section and discussed accordingly.

We thank the Reviewer for this suggestion. For brevity, as previously mentioned, we have opted not to separate the discussion of canopy effects with respect to the observed OF and TF deposition. The measurements are meant to demonstrate the extended application of these samplers, not to study canopy-precipitation interactions in detail. This is the subject of a future manuscript and beyond the scope of this work, as we have noted in the revised Conclusions and Future Directions (Section 4.0) in a prior response above.

**Reviewer 2**

**Overview:**

The paper outlines the design of an automatic precipitation sampler for off-grid use. It is suitable for measuring pH, conductivity, and dissolved organic carbon. The new sampler was tested in the Newfoundland and Labrador Boreal Ecosystem Latitudinal Transect over a two-year period for open-fall and throughfall precipitation. A notable disadvantage to this sampler is the inability to collect snowfall, limiting year-round monitoring of precipitation.

We understand the Reviewer's concern regarding the inability to sample snowfall. We share the same opinion. In fact, since the first iteration, we have modified the funnels and rain sensor chute by installing heating cables to detect, melt, and therefore sample snowfall. This requires a constant power demand in excess of 4 A and, therefore, access to grid power. Since this alteration of the system is still in the testing/development phase, and is not applicable to off-grid sampling, we have decided to exclude the modifications from the current manuscript. This is a substantial engineering challenge and would also undermine our desire to communicate on how to obtain a widely accessible automated deposition sampling method (i.e., cheap).

**General comments**

This work is of great interest to atmospheric scientists and is within the scope of the journal. Overall, this work is appropriate for acceptance in AMT following the revisions outlined below.

The revisions mostly focus on reformatting, reducing lengthy descriptions, and improving clarity (particularly in the introduction and results/discussion sections).

We thank the Reviewer for the positive feedback and appreciate the time they have taken to provide these comments. Please see below where we address these points in our responses and manuscript alterations. Where appropriate, to retain concision, we refer the Reviewer to prior responses above where overlapping concerns with the other Reviewer occur.

**Specific comments**

**Introduction:**

Overall, the introduction is very lengthy. Finding ways to pare down this material would allow for clarity of the important topics related to this work.

We agree with the Reviewer. It is challenging to speak to a broad audience while also retaining a focused Introduction. We appreciate the thoughtfulness of the Reviewer in their suggestions that follow and have done our best to reduce the content of the Introduction. The other Reviewer made a similar suggestion and we have managed to reduce the word count of the Introduction from 2,261 words to 1,783 words in the revision. We hope that this is satisfactory.

Line 74-91: It seems like this paragraph is trying to introduce pH, conductivity, and DOC, however it is hard to separate the information of these three topics from each other. The discussion of pH, conductivity, and DOC are mixed together, making it hard for the reader to parse out the relevant information for each topic. This paragraph needs to be revised and simplified.

We agree with the Reviewer. Given the established measurements of pH and conductivity, we feel that removing the majority of this paragraph to better focus on the deposition collectors themselves would address this comment accordingly as well as the other comments pertaining to the dense and general nature of the Introduction.

Line 119-132: This paragraph discusses persistent organic pollutants (POPs), which the authors do not monitor during the testing period presented in this work. This paragraph could be removed from the introduction and incorporated later as a future use for the sampler.

Line 152-165: This paragraph discusses monitoring reactive nitrogen in atmospheric deposition, however this was not a focus of the precipitation characteristics (pH, conductivity, and DOC) that was highlighted in the results. If this does in fact connect with those three characteristics, those connections need to be made clearer, or this paragraph can be removed.

Response to comments pertaining to L119-132 and L152-165:

We thank the Reviewer for these comments and Reviewer 1 raised similar concerns. The Introduction has been refocused and the mention/discussion of studying the deposition of various chemical species (POPs, ON, etc.) not specifically studied in this work has been either removed completely or integrated into a succinct addition in the revised Conclusions and Future Directions (Section 4.0).

**Methods**

**2.1.2 Heated Precipitation Sensor**

Line 305-306: Is the conductivity that triggers the sensor typically for that of precipitation (both in your geographic region and others)? I wonder about the variability of precipitation globally and if this would cause differences in sampling. In addition, what is the time delay for the lid opening once triggered?

We thank the Reviewer for this comment and Reviewer 1 raised similar concerns. While we acknowledge that conductivity in precipitation could vary depending on sampling locations, the threshold that we report represents the lower limit of the range and this excludes our modification to its design to increase the sensitivity of the sensor by adding the chute. We do not expect precipitation conductance to fall below our threshold unless there is a washout event in the atmosphere the effectively consists of ultrapure water. We note that established sensors on deposition samplers used by government monitoring programs that rely on conductivity would be similarly impacted. As such, we have retained our original lower limit conductivity definition and added the equivalent concentration in sodium chloride. We think this clarifies the operation of the samplers, their detection limits, and hope the Reviewer agrees.

We have added the following to Section 2.1.2 Heated Precipitation Sensor:

The detection of rain modulates the opening and closing of the collection units by an interdigitated resistive sensor (M152; Kemo Electronic GmbH, Geestland, Germany; Figures S6 to S8). **This approach is consistent with established precipitation detection techniques used by government monitoring programs (e.g., CAPMoN; Canadian Air and Precipitation Monitoring Network, 1985a, 1985b).** The rain sensor detects conductive deposition by the completion of a conductive circuit when electrolytes bridge the connection between the interdigitated gold electrodes. The sensor is supplied with 12 VDC from the power system to trigger a relay when precipitation conductance above 1 MΩ·cm conductivity is detected (determined experimentally, see Section S1). **This is equivalent to approximately 8 µM sodium chloride. The sensor detection limit reflects an upper limit of precipitation ion loading because the design of the rain chute leads to an increase in surface area of more than a factor of 25 on which solutes can accumulate to enhance the ionic content of the deposited water.** An output of 12 VDC is sent to the digital control board by the relay when rain is sensed, or 0 VDC in its absence, for signal processing and motor control (Figure S7).

Additionally, Reviewer 1 raised the point regarding the opening and closing times of the lid. We thank the Reviewer for this comment, and we understand the need to communicate that the samplers open rapidly when rain is sensed, for the facts they mention here. The opening of the lid is fast (<5 seconds) and is dependent on the rotation rate of the motor selected. We typically use 2 to 6 rpm motors, depending on their availability from our suppliers. We hope the inclusion of this information in our revision is satisfactory.

An output of 12 VDC is sent to the digital control board by the relay when rain is sensed, or 0 VDC in its absence, for signal processing and motor control (Figure S7). **When rain is sensed, the lid of each sampler in the array is simultaneously opened (<5 seconds) and is dependent on the rotational rate of the lid motor.** To increase the sensitivity of this sensor and to extend the sampling duration when conductive atmospheric constituents are completely washed out of the atmosphere

**Results and Discussion**

**3.1. General design advantage**

Overall, I think the results and discussion could be reformatted. It was confusing to read and keep track of the sampler's validation results and how you were actually applying the sampler to gain new information (like TF and OF comparisons). I appreciate the "General Design Advantages" section to highlight the ability to use this sampler in remote locations and to collect replicates.

We thank the Reviewer for recognizing our rationale for including the "General Design Advantage" section in the manuscript. This is contrary to the comments made by Reviewer 1, as they felt that this section resembles a promotional brochure or technical document. Given the conflicting opinions of the Reviewers, we have elected to retain our preference of having this section in the manuscript.

Below, we have attempted to address the Reviewer's detailed comments on improving the Results and Discussion section of the manuscript. We hope they find these to be acceptable.

**3.3. Comparison of Sample Collection Volumes**

Line 681: If there is a lot of data missing from the ECCC monitoring site, why would you choose this dataset for your comparisons?

We thank the Reviewer for their comment. The Authors initially believed that although a lot of data was missing from the monitoring site in question, it was still better to include any available data for collocated sampling sites. Upon reflection, we now agree that this is not a suitable comparison to include within the main text but is instructive information to include in the

Supporting Information. In also addressing comments made by Reviewer 1, suggesting that our comparison of total deposition with ECCC and DAYMET data is tangential to the central topics of this paper, Section 3.3 was greatly simplified by moving the ECCC/DAYMET comparison to the Supporting Information (Section S2). We agree with both Reviewers that while this was instructive for our biogeochemistry and hydrology work at the NL-BELT, that it is not instructive for the wider scientific community.

Line 689-690: The R2 values are presented in a confusing manner, especially the ones found in parentheses. Please make it more clear which value corresponds to which site.

We thank the Reviewer for their comment. As mentioned above, to address comments made by both Reviewers, Section 3.3. was greatly simplified by moving the ECCC/DAYMET comparison to the Supporting Information. Regarding the $R^2$ values, the way in which they are presented in the Supporting Information has been revised for simplicity as follows:

**S2. Deposition Comparison: This Work, DAYMET, and ECCC**

The DAYMET observations are representative of a larger spatial scale, where our discrete samplers could be subject to heterogeneity in deposition (e.g., orographic precipitation, driven by topography like steep slopes) or impacted by meteorological conditions not captured by the model (e.g., undercatch driven by local winds). The temporally resolved volume comparisons at sampling interval timescales better-demonstrates comparability, despite the systematic differences. **The month-to-month relationships between DAYMET and our observations, as well as between ECCC and our observations, all show strong correlations with linear regressions having $R^2$ values, from north to south, of 0.72, 0.99, 0.99, and 0.86, and N/A, 0.94, N/A, and 0.93, respectively (Table S3).** The discrepancy between DAYMET, ECCC, and our observations for total deposition were highest in the most northerly site, where the experimental site was located on a steep slope, with only 43 % of the predicted volume collected. At all of the sample collection sites on the island of Newfoundland, a consistent difference was observed with 65 ± 4 % of the estimated volume collected, except at GC where our measurements and those from ECCC are identical and starkly contrasting to DAYMET.

Caption of Table S3:

**Table S3.** Collected precipitation volumes from NL-BELT in bulk deposition samplers for rainwater were deployed for one to two months, while snow was collected as an integrated sample throughout the winter because sites were not accessible. The Environment and Climate Change Canada (ECCC) precipitation data was obtained for identical sampling intervals. The DAYMET model for North America (1 km x 1 km resolution) for precipitation was obtained for the identical sampling intervals, which utilizes interpolated and extrapolated data from daily weather observations to predict inputs at the NL-BELT locations. Linear regression results for slope (m) and correlation coefficient ($R^2$) between observations and DAYMET (*italics*), and observations and ECCC (where possible; **underlined**), were calculated. For sampling periods where a measurement was compromised or not collected for a given interval in this work, these are reported as with '-' and an estimated volume from the regression relationship with ECCC is reported in parentheses when used to replace compromised samples.

Overall, Section 3.3 Comparison of Sample Collection Volumes is very long and feels repetitive. This section should be made more concise.

We thank the Reviewer for their comment and appreciate their suggestions. We have moved the ECCC/DAYMET comparison to the Supporting Information, resulting in a more succinct Section 3.3 (1,893 words now reduced to 1,675).

**3.4. Characterizing Chemical Parameters from NL-BELT**

Line 815-819: Mentioning and comparing the pH of the TF samples to the pH of the soil in that area seems like extraneous information. Why are you making this comparison? If it's truly needed, provide some justification or explanation as to why this is important.

We thank the Reviewer for their comment, and we agree that this is extraneous information to include here, as was the comparison in the prior section. Again, while there is value in this for our ongoing work at the NL-BELT, we agree that it was a misplaced idea on our part to include it in the main text of manuscript.

We have removed this material entirely from Section 3.4.1 and incorporated it into Table 1 as "footnote a" for contextual purposes:

[a]Soil pH for the organic and mineral soil horizons determined by addition of 400 µL of aqueous 0.5 M $CaCl_2$ to a 50:50 w/w slurry of dried soil in deionised water. **Note: the four remote NL-BELT sites are dominated by balsam fir trees underlain by humo-ferric podzol soil with pH ranging between 3.0 and 4.5.**

Section 3.4.2 – Most other sections have a comparison of OF and TF samples, but this one does not. Was that intentional? A brief statement comparing these two would be great.

We thank the Reviewer for their comment and this brief statement has been included within Section 3.4.2:

In all the collected OF and TF precipitation samples, across all four NL-BELT sites, the average measured conductivity values ranged from 21 to 166 µS/cm **following no apparent seasonal or temporal trend** (Figure 6). **Additionally, the conductivity in both OF and TF also appear to vary across the field sites - only within the 2016 TF samples does the conductivity**

**appear to increase with decreasing latitude.** Yet, with the typical conductivity of surface and drinking waters being between 1 to 1000 µS/cm (Lin et al., 2017), and typically below 200 µS/cm in stream water measurements within the watersheds of each of the NL-BELT sites, our observations are comparable and fall within the expected range. Our field blanks – encompassing a variety of materials and apparatuses, and our cleaning procedures, routinely produced conductivities of $9 \pm 5$ µS/cm.

Line 893-894: p-values to support your statements that the conductivity of HgCl2 in water is comparable to field blanks and less than your samples would strengthen this argument.

The Reviewer is making a request that may not be as informative as they wish it to be, but we have decided to incorporate the results of a statistical comparison in line with their request. Our measurements of $HgCl_2$ in water at the saturation limit is an upper boundary. This is the measured value of $13.6 \pm 0.4$ µS/cm is expected to be much higher than its impact on any sample where the small volume of such solutions (1 to 5 mL) is diluted by the total volume of collected precipitation. Thus, the comparison and resulting statistics are expected to be skewed towards similarity as they do not account for this physical reality.

In addition, our field blanks show less conductive contamination compared to this upper limit of saturated $HgCl_2$ statistically, and the samples are significantly higher ($p = 7.65 \times 10^{-8}$; unpaired t-test). Therefore, the results hold up in line with expectations of high-quality analytical performance, despite the above stated caveat. We have added brief statistical outcomes from unpaired t tests to the main manuscript:

Our field blanks – encompassing a variety of materials and apparatuses, and our cleaning procedures, routinely produced conductivities of $9 \pm 5$ µS/cm. The conductivity of saturated $HgCl_2$ in water (at 0.1% vol/vol) was $13.6 \pm 0.4$ µS/cm, which is also comparable to **but statistically higher than** our field blanks **(p = 0.0015; unpaired t test)** and less than what was observed for our samples **($p < 2 \times 10^{-6}$ for each site considered separately and also across all sites; unpaired t-test).**

Line 958-963: You mention wildfire plumes being a potential cause of increased DOC levels in precipitation. Was there a wildfire event near your sites during your sampling period to warrant mentioning this possibility? Or would the possibility of increased DOC still be observed some amount of time after a wildfire event? In order to justify including this argument, I would like to see if you could make a potential correlation to an actual event that may explain this.

We thank the Reviewer for highlighting the very important link between wildfire plumes and increased DOC deposition. The province of Newfoundland and Labrador does experience wildfires, forest fire season is in effect from May to September each year. Based on 20 years of wildfire data, there are on average 118 wildfires burning 22,993 hectares in Newfoundland and Labrador each year (Government of Newfoundland and Labrador, 2023). We strongly believe this argument is warranted since (i) there has been an increase in wildfire activity across North America and (ii) several studies have now linked increased organic and inorganic carbon deposition to wildfire events. (Wagner et al., 2021; Coward et al., 2022; Barton et al., 2024). Given the variability in atmospheric transport from these fire locations (typically in Labrador, but also reaching Quebec and further west), it is challenging to provide deep insight regarding the magnitude of impact in monthly integrated samples without detailed back trajectory analysis driven by known precipitation intercepting these airmasses. So, although we did not make a quantitative determination of wildfire plumes during the campaign, it would be unusual not to include wildfires as a possible reason for increased DOC deposition.

The discussion on DOC seems less than a validation of instrument performance, but rather a capability of the sampler. Unless you have DOC data from your sampling locations to compare to, this seems like new information for remote sampling sites.

We thank the Reviewer for this comment. We may have miscommunicated our approach and would now like to clarify it. In Section 3.4, we aimed to highlight the sampler's capability to accurately quantify precipitation pH, conductivity, and DOC fluxes, with a special interest in demonstrating the capacity to discern and investigate canopy effects. To validate our new DOC measurements, we compared our observed fluxes to other studies in forested regions. We strongly feel that this section is a validation of the instrument performance and re-emphasize that future applications for these samplers are stated in the Conclusions and Future Directions (Section 4.0), as further scientific inquiry is beyond the scope of this work.

To add clarity to the manuscript, we have added the following text in Section 3.4.3 (paragraph following Figure 7):

Additionally, we cannot rule out that the chemical speciation differs between OF, TF, and SF even if the DOC values are similar, but such insights require more selective instrumentation for chemical analysis (e.g., high resolution mass spectrometry).

**The ability to accurately determine DOC in OF and TF precipitation demonstrates the capability of the automated deposition samplers. To validate our measurements, we compared our observed fluxes to other studies in different forest types.** Mean annual DOC fluxes were generally similar to those reported in some other boreal forests (Table 3).

Line 1014: what evidence do you have that supports the same that the automated system better maintains the integrity of DOC in samples?

We thank the Reviewer for highlighting this and we welcome the opportunity to clarify our approach. Our rationale in stating that the automated system maintains the integrity of DOC in the collected samples is because we have followed the standard approach used by biogeochemists to study DOC in soil and freshwater samples, by microbially fixing them through the addition of

HgCl₂. In addition, we have outlined apparatus cleaning procedures to minimize contamination. Finally, the sampler design also prevents the intrusion of forest litter and other materials that could potentially influence the levels of DOC in forested regions if these were to be submerged in the collected aqueous precipitation sample, leading to bias due to leaching of organic compounds from these solid organic materials. This is particularly important if the objective is to selectively quantify DOC in wet deposition.

To better address this comment, we have made a small addition to the final paragraph of Section 3.4.3 which now reads:

The automated system better maintains the integrity of DOC in the samples. **This was achieved by following standard procedures for biogeochemical sample preservation (i.e., adding HgCl₂) (Argentino et al, 2023), employing a rigorous cleaning procedure, and preventative design against the intrusion of forest litter** which could result in a positive bias for DOC in the collected precipitation.

Additionally, we've added a new subsection "Sample Perseveration" (2.3.1) to address comments regarding topics such as sample preservation and our approach to method validation.

Technical Corrections Starting with section 3.4.3 – it is mentioned that flux (with the units: mg C m-2 d -1 ) is used. However, at several points in this discussion flux is given as mg C m-2 a -1 . I'm providing a list locations I've found this mistake, but please check throughout the manuscript for places I may have missed:
      Lines 50, 921, 925, 944, 955, 966, 997, 998, 100

We thank the Reviewer for this comment; however, we think there was some miscommunication with our approach and we would like the opportunity to clarify. The deposition fluxes were calculated daily; however, this was summed for each sampling period and reported as an equivalent annual flux with units of (mg m$^{-2}$ a$^{-1}$). We have checked the values and units throughout the manuscript and Supporting Information to confirm that they are correct in all locations.

To clarify, the text at the end of the first paragraph of Section 3.4.3 now reads:

The concentrations were converted to elemental fluxes using the volume of precipitation collected, the area of the funnel and the number of sampling days of each sampling period (Figure 7). The total flux for each sample period was summed **and reported as an equivalent annual flux with the following units: mg m$^{-2}$ a$^{-1}$. Annual fluxes** ranged from 600 to 4200 mg C m$^{-2}$ a$^{-1}$ across the study sites for the snow free period (Table S4).

[revised manuscript text omitted]